# Refining Adaptive Zeroth-Order Optimization at Ease

**Yao Shu** [1]  **Qixin Zhang** [2]  **Kun He** [3]  **Zhongxiang Dai** [4]

## Abstract

Recently, zeroth-order (ZO) optimization plays an essential role in scenarios where gradient information is inaccessible or unaffordable, such as black-box systems and resource-constrained environments. While existing adaptive methods such as ZO-AdaMM have shown promise, they are fundamentally limited by their underutilization of moment information during optimization, usually resulting in underperforming convergence. To overcome these limitations, this paper introduces *Refined Adaptive Zeroth-Order Optimization* ($\mathcal{R}$-AdaZO). Specifically, we first show the untapped variance reduction effect of first moment estimate on ZO gradient estimation, which improves the accuracy and stability of ZO updates. We then refine the second moment estimate based on these variance-reduced gradient estimates to better capture the geometry of the optimization landscape, enabling a more effective scaling of ZO updates. We present rigorous theoretical analysis to show **(I)** *the first analysis* to the variance reduction of first moment estimate in ZO optimization, **(II)** *the improved second moment estimates* with a more accurate approximation of its variance-free ideal, **(III)** *the first variance-aware convergence framework* for adaptive ZO optimizers, which may be of independent interest, and **(IV)** *the faster convergence* of $\mathcal{R}$-AdaZO than existing baselines like ZO-AdaMM. Our extensive experiments, including synthetic problems, black-box adversarial attack, and memory-efficient fine-tuning of large language models (LLMs), further verify the superior convergence of $\mathcal{R}$-AdaZO, indicating that $\mathcal{R}$-AdaZO offers an improved solution for real-world ZO optimization challenges.

## 1. Introduction

Zeroth-order (ZO) optimization has emerged as an indispensable technique at the forefront of machine learning, addressing critical challenges where gradient information is either unavailable or computationally prohibitive. This necessity stems from the prevalence of black-box optimization problems, such as adversarial attacks (Ru et al., 2020; Hiranandani et al., 2021), and resource-constrained environments, like fine-tuning large language models (LLMs) on memory-limited devices (Malladi et al., 2023; Zhang et al., 2024b). Consequently, ZO optimization algorithms, which rely solely on function evaluations, have become a crucial alternative to traditional gradient-based methods. Despite the growing body of research in ZO optimization, a significant portion of existing methods adapt stochastic gradient descent (SGD) updates to the ZO setting (Liu et al., 2018a;b; Shu et al., 2023; 2024). This reliance on SGD, however, will lead to performance limitations, especially in complex and non-convex optimization landscapes. The need for more adaptive and versatile update mechanisms is hence evident. However, the exploration of adaptive strategies beyond SGD-based updates remains surprisingly limited.

While adaptive methods such as ZO-AdaMM (Chen et al., 2019; Nazari et al., 2020) have demonstrated potential in addressing the missing adaptivity in zeroth-order optimization, they are fundamentally limited by their underutilization of moment information, often resulting in suboptimal convergence rates. This limitation in fact arises from their reliance on noisy and high-variance gradient estimates derived solely from function evaluations—a stark contrast to the first-order (FO) methods that leverage direct and more stable gradients. This issue becomes even more pronounced in high-dimensional and complex settings.

To address this critical limitation, we introduce *Refined Adaptive Zeroth-Order Optimization* ($\mathcal{R}$-AdaZO), a novel approach that effectively capitalizes on moment information through two key innovations. First, $\mathcal{R}$-AdaZO is the first to analyze the untapped but inherent variance reduction effect of the first moment estimates on the gradient estimates in ZO optimization, leading to more accurate and stable ZO updates. This is accomplished through the integration of historical gradient estimates, which effectively averages out the estimation noise (Sec. 4.1). Second, $\mathcal{R}$-AdaZO refines

[1]Hong Kong University of Science and Technology (Guangzhou) [2]Nanyang Technological University [3]Huazhong University of Science and Technology [4]The Chinese University of Hong Kong, Shenzhen. Correspondence to: Zhongxiang Dai <daizhongxiang@cuhk.edu.cn>.

*Proceedings of the 42nd International Conference on Machine Learning*, Vancouver, Canada. PMLR 267, 2025. Copyright 2025 by the author(s).

the second moment using these variance-reduced gradient estimates, enabling better adaptation to the underlying geometry of the optimization landscape and facilitating a more effective scaling of ZO updates (Sec. 4.2).

Beyond simply presenting $\mathcal{R}$-AdaZO, we provide a thorough analysis that combines rigorous theoretical guarantees with extensive empirical validation, demonstrating its effectiveness. Specifically, we first provide the assumptions used in our theoretical analysis (Sec. 5.1). We then theoretically analyze that incorporating first-moment estimates into ZO optimization significantly reduces the variance, leading to more stable and reliable ZO updates, and theoretically demonstrate that our refined second moment estimates provide a more accurate approximation of its corresponding variance-free ideal (Sec. 5.2). We further introduce the first variance-aware framework to prove the convergence of adaptive ZO optimization methods, which is not limited to our specific method and can be used to analyze a wider range of similar algorithms, and theoretically prove that $\mathcal{R}$-AdaZO converges faster than established baseline methods, such as ZO-AdaMM, demonstrating its efficiency in optimization (Sec. 5.3). Through extensive experiments, including synthetic problems (Sec. 6.1), black-box adversarial attack (Sec. 6.2), and memory-efficient LLM fine-tuning (Sec. 6.3), we demonstrate that $\mathcal{R}$-AdaZO consistently outperforms existing methods in practice, exhibiting superior convergence.

To summarize, our contributions in this work include:

- We propose $\mathcal{R}$-AdaZO to enhance the utilization of moment information in ZO optimization and significantly improve the convergence of adaptive ZO optimizers.
- We theoretically show **(I)** *the first analysis* to the variance reduction of first moment estimates in ZO optimization, **(II)** *the effects of our refined second moment estimates*, **(III)** *the first variance-aware convergence framework* for adaptive ZO methods, which may be of independent interest, and **(IV)** *the improved convergence* of $\mathcal{R}$-AdaZO.
- We use extensive empirical validation to show the consistent performance gains of $\mathcal{R}$-AdaZO over baselines.

## 2. Related Work

Recent ZO optimization research focuses on two key areas: ZO gradient estimation and ZO update rules.

**ZO Gradient Estimation.** Since ZO optimization only relies on function values, gradient estimation is essential for effective optimization. A common approach is to use finite difference approximations under input perturbations. Nesterov & Spokoiny (2017) propose to use Gaussian random noise perturbations, demonstrating theoretical convergence with smooth perturbations. Other methods also propose to use uniform sampling from the unit sphere (Flaxman et al.,

2005) or coordinate-wise perturbations (Lian et al., 2016). These methods often have a noisy gradient estimation. To address this, Cheng et al. (2021) introduce prior-guided gradient estimation, which leverages previous estimates to improve the current one, effectively smoothing the estimation noise. Recently, Shu et al. (2023; 2024) propose using kernel methods to learn a surrogate model of the objective function from historical function values, allowing for more accurate gradient estimation. Another line of work has focused on linear interpolation strategies for more accurate estimates by reusing queries from prior iterations to reduce complexity while maintaining sample quality (Wang et al., 2024b). Note that this paper does not aim to introduce a new gradient estimation approach, but focus on developing advanced update rules that are applicable to all these existing estimation methods.

**ZO Update Rules.** Building upon the estimated gradients from various ZO estimation methods, ZO optimizers often directly adopt update rules from first-order (FO) optimization. E.g., a large portion of existing ZO optimizers use stochastic gradient descent (SGD) and its variants as their update mechanism (Ghadimi & Lan, 2013; Ghadimi et al., 2016; Nesterov & Spokoiny, 2017; Liu et al., 2018b;a; 2019; Cheng et al., 2021; Shu et al., 2023). While simple to apply, the slow convergence of SGD has motivated few efforts (Chen et al., 2019; Nazari et al., 2020; Jiang et al., 2024) to explore the use of adaptive methods, such as Adam (Kingma & Ba, 2015), as the ZO update rule. However, these attempts often under-utilize the moment information inherent in adaptive methods when applied to ZO optimization, leading to suboptimal convergence. This paper addresses this critical issue by proposing refined update rules that are specifically designed to better leverage moment information, ultimately leading to more efficient ZO optimization.

## 3. Background

This paper tackles a stochastic zeroth-order (ZO) optimization problem, aiming to minimize the expected value of a function, defined as:

$$\min_{\boldsymbol{\theta} \in \mathbb{R}^d} F(\boldsymbol{\theta}) \triangleq \mathbb{E}_\xi \left[ f(\boldsymbol{\theta}; \xi) \right] \tag{1}$$

where $\boldsymbol{\theta} \in \mathbb{R}^d$ and $f(\boldsymbol{\theta}; \xi)$ is a scalar-valued function whose evaluation depends on the parameters $\boldsymbol{\theta}$ and a random variable $\xi$ sampled from an underlying distribution. Crucially, we have access only to function evaluations $f(\boldsymbol{\theta}; \xi)$ and not its gradient $\nabla_{\boldsymbol{\theta}} f(\boldsymbol{\theta}; \xi)$. Throughout this paper, we adopt the following notational conventions. Vectors are represented in boldface, e.g., $\boldsymbol{\theta}$, and scalar constants are denoted by uppercase letters, e.g., $L$. All vector operations are assumed to be element-wise unless explicitly stated otherwise. We denote by $\nabla_i F$ the partial derivative of function $F$ with respect to

the $i$-th coordinate.

**ZO Gradient Estimation.** In ZO optimization, the absence of direct access to gradients, denoted as $\nabla_{\boldsymbol{\theta}} f(\boldsymbol{\theta}; \xi)$, necessitates the use of gradient estimation techniques that rely solely on function evaluations. A widely used method is to approximate gradients using finite differences. E.g., let random vectors $\{\boldsymbol{u}_k\}_{k=1}^K$ be drawn uniformly from the sphere $\mathbb{S}^{d-1}$ of a unit ball $\mathbb{B}^d$, a common ZO gradient estimator, which is used throughout this paper, can be formed as:

$$\hat{\nabla} f(\boldsymbol{\theta}, \xi) \triangleq \frac{d}{K} \sum_{k=1}^K \frac{f(\boldsymbol{\theta} + \mu \boldsymbol{u}_k; \xi) - f(\boldsymbol{\theta}; \xi)}{\mu} \boldsymbol{u}_k \quad (2)$$

where $\mu > 0$ is a smoothing parameter, and $K$ is the number of random vectors. While this paper utilizes this specific ZO gradient estimator as its foundation, the proposed method is extensible to other ZO gradient estimators as well.

**Adaptive ZO Optimization.** ZO optimization methods with a fixed step size typically suffer from slow convergence. To address this, adaptive methods like ZO-AdaMM (Chen et al., 2019) are used, which incorporate momentum using first moment estimates and per-parameter learning rates using second moment estimates. Specifically, in ZO-AdaMM, the parameter updates are computed as follows for every iteration $t$ (see also Algo. 1):

$$\boldsymbol{m}_t \leftarrow \beta_1 \boldsymbol{m}_{t-1} + (1 - \beta_1) \boldsymbol{g}_t \quad \text{(First Moment Est.)}$$
$$\boldsymbol{v}_t \leftarrow \beta_2 \boldsymbol{v}_{t-1} + (1 - \beta_2) \boldsymbol{g}_t^2 \quad \text{(Second Moment Est.)}$$
$$\boldsymbol{\theta}_t \leftarrow \boldsymbol{\theta}_{t-1} - \eta \frac{\boldsymbol{m}_t}{\sqrt{\boldsymbol{v}_t + \zeta}} \quad \text{(ZO Update)}$$
$$(3)$$

where $\boldsymbol{g}_t = \hat{\nabla} f(\boldsymbol{\theta}_{t-1})$ defined in (2), $\beta_1, \beta_2 \in (0, 1)$ are exponential decay rates for moment estimates, and $\zeta$ is a small constant to prevent dividing by zero.

However, while these adaptive ZO approaches have shown promise, they often underutilize the moment information in the context of ZO optimization: *(a)* They typically treat first moment estimate $\boldsymbol{m}_t$ as standard velocity accumulation in FO optimization, failing to consider its underlying variance reduction effect in ZO optimization by accumulating information from previous gradient estimates. *(b)* They fail to apply this variance-reduced gradient estimates to refine the second moment estimate $\boldsymbol{v}_t$, causing a less effective scaling of ZO updates.

## 4. Refined Adaptive ZO Optimization

To address the underutilization of momentum information in existing adaptive ZO optimization methods, we introduce $\mathcal{R}$-AdaZO (*Refined Adaptive Zeroth-Order Optimization*). Specifically, we first analyze the untapped variance reduction effect of first moment estimates on ZO gradient estimation, which is important for accurate and stable ZO

---

**Algorithm 1** ZO-AdaMM

**Input:** $\beta_1, \beta_2, \zeta, \eta, f$
**Initialize:** $\boldsymbol{\theta}_0, \boldsymbol{m}_0, \boldsymbol{v}_0$
**for** *iteration* $t \in [T]$ **do**
    $\boldsymbol{g}_t \leftarrow \hat{\nabla} f(\boldsymbol{\theta}_{t-1}, \xi_t)$
    $\boldsymbol{m}_t \leftarrow \beta_1 \boldsymbol{m}_{t-1} + (1 - \beta_1) \boldsymbol{g}_t$
    $\boldsymbol{v}_t \leftarrow \beta_2 \boldsymbol{v}_{t-1} + (1 - \beta_2) \boldsymbol{g}_t^2$
    $\boldsymbol{\theta}_t \leftarrow \boldsymbol{\theta}_{t-1} - \eta \frac{\boldsymbol{m}_t}{\sqrt{\boldsymbol{v}_t + \zeta}}$
**Output:** $\boldsymbol{\theta}_T$

---

**Algorithm 2** $\mathcal{R}$-AdaZO

**Input:** $\beta_1, \beta_2, \zeta, \eta, f$
**Initialize:** $\boldsymbol{\theta}_0, \boldsymbol{m}_0, \boldsymbol{v}_0$
**for** *iteration* $t \in [T]$ **do**
    $\boldsymbol{g}_t \leftarrow \hat{\nabla} f(\boldsymbol{\theta}_{t-1}, \xi_t)$
    $\boldsymbol{m}_t \leftarrow \beta_1 \boldsymbol{m}_{t-1} + (1 - \beta_1) \boldsymbol{g}_t$
    $\boldsymbol{v}_t \leftarrow \beta_2 \boldsymbol{v}_{t-1} + (1 - \beta_2) \boldsymbol{m}_t^2$
    $\boldsymbol{\theta}_t \leftarrow \boldsymbol{\theta}_{t-1} - \eta \frac{\boldsymbol{m}_t}{\sqrt{\boldsymbol{v}_t + \zeta}}$
**Output:** $\boldsymbol{\theta}_T$

---

updates (Sec. 4.1). We then apply these variance-reduced estimates to construct a refined second moment, enabling more effective scaling of ZO updates (Sec. 4.2).

### 4.1. Variance Reduction in First Moment Estimates

First moment estimation, while conventionally used for convergence speedup, inherently serve as a variance reduction mechanism for noisy gradients. To show this, consider the following standard first moment estimate with $\beta_1 \in (0, 1)$:

$$\boldsymbol{m}_t \leftarrow \beta_1 \boldsymbol{m}_{t-1} + (1 - \beta_1) \boldsymbol{g}_t \quad (4)$$

where $\boldsymbol{m}_t$ is the estimated first moment at iteration $t$ and $\boldsymbol{g}_t$ is the gradient estimate at $\boldsymbol{\theta}_{t-1}$ via (2). Intuitively, this update averages the current noisy gradient estimate with past, correlated estimates. This averaging process effectively smooths out noise in gradient estimates, thereby reducing variance. For example, averaging two independent noisy gradient estimates (ie, $\boldsymbol{m}_{t-1}$ and $\boldsymbol{g}_t$) of variance $\sigma^2$ results in a variance of $[\beta_1^2 + (1 - \beta_1)^2]\sigma^2$, which is less than $\sigma^2$. While current and past gradient estimates are not fully independent in practice, their local correlation still enables variance reduction through this averaging, which we will show theoretically in Sec. 5.

While this variance reduction effect has been proven in FO optimization (Liu et al., 2020), it is significantly more crucial in ZO optimization. Unlike FO methods that compute gradients directly with relatively low variance, ZO optimization approximates gradients using function evaluations (as in (2)), resulting in inherently noisier estimates. This disparity underscores the critical importance of the variance

reduction effect of first moment estimates in ZO optimization, a connection we are the first to identify. We further provide theoretical support for this in Sec. 5.

### 4.2. Refinement to Second Moment Estimates

The second key innovation of $\mathcal{R}$-AdaZO lies in its refined second moment estimate, which is crucial for the adaptivity in ZO optimization. Existing adaptive ZO methods (Chen et al., 2019; Nazari et al., 2020) update the second moment estimate directly using the squared noisy gradient estimates:

$$\boldsymbol{v}_t = \beta_2 \boldsymbol{v}_{t-1} + (1 - \beta_2)\boldsymbol{g}_t^2 \ . \tag{5}$$

However, this approach can be suboptimal in the ZO setting owing to the inherent high variance of the gradient estimates in (2), which could lead to unstable and unreliable second moment estimates. We thus propose to address this issue by simply leveraging the variance-reduced gradient information from the first moment. That is, we update the second moment estimate as below, which interestingly shares similar form with RMSProp (Hinton, 2012).

$$\boldsymbol{v}_t = \beta_2 \boldsymbol{v}_{t-1} + (1 - \beta_2)\boldsymbol{m}_t^2 \ . \tag{6}$$

The first moment estimate, as revealed in Sec. 4.1, acts as a variance reduction mechanism by averaging historical gradient information. Using the squared first moment estimate then probably provides a smoothed and more stable second moment estimate. This refinement therefore may enable a more accurate representation for the underlying geometry of the optimization landscape, resulting in more effective scaling of ZO updates and thus accelerated convergence. Specifically, consider a scenario where $\mathbb{E}[\boldsymbol{m}_t] = \mathbb{E}[\boldsymbol{g}_t]$ but $\boldsymbol{m}_t$ has significantly lower variance than $\boldsymbol{g}_t$ due to the smoothing effect, given the same $\boldsymbol{v}_{t-1}$, we can see that the refined $\boldsymbol{v}_t$ in (6) achieves a smaller expected value compared to the standard one in (5). Hence, the update step (see (7)) using this refined $\boldsymbol{v}_t$ in (6) is likely to be larger, allowing the algorithm to move faster towards the optimum. This claim will be rigorously established in Sec. 5.

### 4.3. Final Algorithm

Given the first and second moment estimates in (4) and (6) respectively, $\mathcal{R}$-AdaZO updates parameters by:

$$\boldsymbol{\theta}_t = \boldsymbol{\theta}_{t-1} - \eta \frac{\boldsymbol{m}_t}{\sqrt{\boldsymbol{v}_t + \zeta}} \tag{7}$$

where $\eta$ is the base learning rate and $\zeta$ is a small constant for numerical stability. This update rule adaptively scales the effective learning rate based on the local geometry while incorporating the variance-reduced gradient estimates. The complete $\mathcal{R}$-AdaZO algorithm is detailed in our Algo. 2.

**Computational and Memory Complexity.** $\mathcal{R}$-AdaZO incurs the same computational cost of $\mathcal{O}(Kd)$ per iteration

for moment estimates and ZO updates (excluding function evaluations), and the same memory footprint of $\mathcal{O}(d)$ as ZO-AdaMM for moment estimates, where $K$ is the number of function evaluations and $d$ is the dimension of parameter $\boldsymbol{\theta}$.

**Ease of Implementation.** A key advantage of $\mathcal{R}$-AdaZO is its simple implementation. The core change involves updating the second moment estimate using the squared first moment estimate, a one-line change for existing adaptive ZO optimizers. This minimal change enables easy integration and fast deployment, while improving convergence.

## 5. Theoretical Analysis

This section presents a theoretical foundation for the efficacy of $\mathcal{R}$-AdaZO. We structure our analysis as follows: First, we introduce the required assumptions and preliminaries (Sec.5.1). Second, we prove the variance reduction in first moment estimate and the improvement of our refined second moment in $\mathcal{R}$-AdaZO (Sec. 5.2). Finally, we present the first variance-aware convergence framework for adaptive ZO methods and demonstrate the improved convergence of $\mathcal{R}$-AdaZO over other baselines (Sec. 5.3). To ease our proof, we primarily provide the theoretical analysis for (2) with $\boldsymbol{u} \sim \text{Unif}(\mathbb{S}^{d-1})$.

### 5.1. Assumptions and Preliminaries

Our theoretical framework is built upon two fundamental assumptions concerning the non-convex function $F$. We impose a bounded function value as well as a coordinate-wise Lipschitz smoothness (Assump. 1), with a bounded variance of function values (Assump. 2). Of note, coordinate-wise Lipschitz smoothness is commonly used in the analysis of FO adaptive gradient methods, e.g., (Zhang et al., 2024a; Wang et al., 2024a).

**Assumption 1.** $\forall \boldsymbol{\theta}, \boldsymbol{\theta}' \in \mathbb{R}^d$ and $\forall i \in [d]$,

$$\left| f(\boldsymbol{\theta}, \xi) \right| \leq C \ , \tag{8}$$

$$\left| \nabla_i F(\boldsymbol{\theta}) - \nabla_i F(\boldsymbol{\theta}') \right| \leq L \left\| \boldsymbol{\theta} - \boldsymbol{\theta}' \right\| \ . \tag{9}$$

**Assumption 2.** $\forall \boldsymbol{\theta} \in \mathbb{R}^d$,

$$\mathbb{E}_\xi \left[ \left| f(\boldsymbol{\theta}, \xi) - F(\boldsymbol{\theta}) \right|^2 \right] \leq \sigma^2 \ . \tag{10}$$

Directly establishing the convergence of $\mathcal{R}$-AdaZO through the function $F$ presents a primary challenge for adaptive ZO methods, due to the bias (i.e., $\mathbb{E}\left[\hat{\nabla} f(\boldsymbol{\theta}, \xi)\right] \neq \nabla F(\boldsymbol{\theta})$) arising from the gradient estimation in (2). Thus, we propose to prove the convergence of $\mathcal{R}$-AdaZO with respect to the randomized smoothing function $F_\mu$ (Duchi et al., 2012) defined in (11) where $\boldsymbol{u}$ is a random vector drawn uniformly from a unit ball $\mathbb{B}^d$ and $\mu > 0$ is a smoothing parameter. Of note, ZO gradient (2) with $\boldsymbol{u} \sim \text{Unif}(\mathbb{S}^{d-1})$ in fact leads to

an unbiased gradient estimation for this smoothing function, which we will show in Lemma 5.1 below.

$$F_\mu(\boldsymbol{\theta}) \triangleq \mathbb{E}_{\boldsymbol{u} \sim \mathbb{B}^d} \left[ F(\boldsymbol{\theta} + \mu \boldsymbol{u}) \right] . \tag{11}$$

We introduce the following Lemma 5.1 (proof in Appx. A.1) to justify why $F_\mu$, instead of $F$, servers as a better choice for the convergence framework of adaptive ZO methods.

**Lemma 5.1.** *Given* (2) *with* $\boldsymbol{u} \sim Unif(\mathbb{S}^{d-1})$, *let Assump. 1 hold,* $\forall \boldsymbol{\theta} \in \mathbb{R}^d$ *and* $\forall i \in [d]$,

$$\mathbb{E}\left[ \hat{\nabla} f(\boldsymbol{\theta}, \xi) \right] = \nabla F_\mu(\boldsymbol{\theta}) , \tag{12}$$

$$\mathbb{E}\left[ \left\| \nabla F(\boldsymbol{\theta}) - \nabla F_\mu(\boldsymbol{\theta}) \right\| \right] \leq \mu L \sqrt{d} . \tag{13}$$

**Remark.** Lemma 5.1 establishes that *(a)* $\nabla F_\mu$ is the expectation of the gradient estimate in (2), thereby overcoming the bias challenge mentioned above, and *(b)* the discrepancy between $\nabla F_\mu$ and $\nabla F$ is bounded above by $\mathcal{O}(\mu)$, implying that the convergence of $\mathcal{R}$-AdaZO with respect to $\nabla F$ can be easily derived after obtaining the results with respect to $\nabla F_\mu$. In light of these, $F_\mu$ will be a good choice for the convergence framework of adaptive ZO methods.

In addition, we provide the following Lemma 5.2 (proof in Appx. A.2) to ease our proof.

**Lemma 5.2.** *Given* (2) *with* $\boldsymbol{u} \sim Unif(\mathbb{S}^{d-1})$, *let Assump. 1 hold,* $\forall \boldsymbol{\theta}, \boldsymbol{\theta}' \in \mathbb{R}^d$ *and* $\forall i \in [d]$,

$$\left| \nabla_i F_\mu(\boldsymbol{\theta}) - \nabla_i F_\mu(\boldsymbol{\theta}') \right| \leq L \left\| \boldsymbol{\theta} - \boldsymbol{\theta}' \right\| , \tag{14}$$

$$\mathbb{E}\left[ \left| \hat{\nabla}_i f(\boldsymbol{\theta}, \xi) - \nabla_i F_\mu(\boldsymbol{\theta}) \right|^2 \right] \leq \frac{8(\sigma^2 + C^2)d}{K\mu^2} . \tag{15}$$

**Remark.** Lemma 5.2 establishes that *(a)* $F_\mu$ exhibits the same Lipschitz smoothness as $F$, and *(b)* the gradient variance associated with ZO optimization can be substantially large, particularly when $K \ll d$ and $\mu$ is small. Therefore, variance reduction is critical for improved ZO optimization.

## 5.2. Analysis on First and Second Moment Estimates

We first theoretically show the underlying variance reduction effect of first moment estimate in (4) using variance $\Sigma^2$ defined below in Thm. 5.3 (proof in Appx. A.3).

$$\Sigma^2 \triangleq \frac{8(\sigma^2 + C^2)d}{K\mu^2} . \tag{16}$$

**Theorem 5.3.** *Given first and second moment estimates* (4) *and* (6) *respectively, with Assump. 1, 2,* $\forall t \geq 1, \forall i \in [d]$,

$$\mathbb{E}\left[ \left| \boldsymbol{m}_{t,i} - \nabla_i F_\mu(\boldsymbol{\theta}_{t-1}) \right|^2 \right] \leq$$

$$\underbrace{\frac{1 - \beta_1}{1 + \beta_1} \Sigma^2}_{\text{Variance}} + \underbrace{\frac{\beta_1 (1 + \beta_1) L^2 \eta^2 d}{(1 - \beta_1)^2 (1 - \beta_2)} + \beta_1^t \mathbb{E}\left[ \left| \nabla_i F_\mu(\boldsymbol{\theta}_{t-1}) \right|^2 \right]}_{\text{Squared Bias}} .$$
$$\tag{17}$$

**Remark.** To the best of our knowledge, this theorem provides the first fundamental variance-bias decomposition for the first moment estimate in adaptive ZO algorithms. The variance, given by $\frac{1-\beta_1}{1+\beta_1}\Sigma^2$, arises from the randomness in gradient estimator (2) and reduces $\Sigma^2$ in (15) by a factor of $\frac{1-\beta_1}{1+\beta_1}$, which can be further improved with a large $\beta_1$. This thus theoretically demonstrates the variance reduction effect of first moment estimate in (4), which goes beyond increasing $K$ to reduce variance. The bias, stemming from the difference between current and past estimates, can be reduced by using a small learning rate $\eta$, which limits the magnitude of update steps, or a small $\beta_1$, which reduces the influence of past estimates. So, this decomposition unveils a fundamental trade-off controlled by the utilization (i.e., $\beta_1$) of past estimates between variance and bias. Particularly, when $\beta_1 = 0$, (17) simplifies to (15).

We then theoretically show that our refined second moment update in (6) is likely to be a more accurate approximation to its variance-free ideal in (18) and hence may better capture the underlying geometry of optimization landscape than (5) used in ZO-AdaMM, with the following Thm. 5.4 (proof in Appx. A.4) and Cor. 5.5 (proof in Appx. A.5).

$$\boldsymbol{v}_{t,i} = \beta_2^t \boldsymbol{v}_{0,i} + \sum_{\tau=1}^t (1 - \beta_2) \beta_2^{t-\tau} \left| \nabla_i F_\mu(\boldsymbol{\theta}_{\tau-1}) \right|^2 . \tag{18}$$

**Theorem 5.4.** *Given second moment estimate* (6), *with Assump. 1, 2,* $\forall t \geq 1$ *and* $\forall i \in [d]$,

$$\mathbb{E}\left[ \boldsymbol{v}_{t,i} \right] \leq \beta_2^t \boldsymbol{v}_{0,i} + (1 - \beta_1)\Sigma^2 + \frac{\beta_1 (1 + \beta_1)^2 L^2 \eta^2 d}{(1 - \beta_1)^2 (1 - \beta_2)} +$$

$$\frac{(1 + \beta_1)^2}{\beta_1} \sum_{\tau=1}^t (1 - \beta_2) \beta_2^{t-\tau} \mathbb{E}\left[ \left| \nabla_i F_\mu(\boldsymbol{\theta}_{\tau-1}) \right|^2 \right] .$$
$$\tag{19}$$

**Corollary 5.5.** *Given second moment estimate in* (5), *with Assump. 1, 2,* $\forall t \geq 1$ *and* $\forall i \in [d]$,

$$\mathbb{E}\left[ \boldsymbol{v}_{t,i} \right] \leq \beta_2^t \boldsymbol{v}_{0,i} + (1 + \beta_1)\Sigma^2 +$$

$$\frac{(1 + \beta_1)^2}{\beta_1} \sum_{\tau=1}^t (1 - \beta_2) \beta_2^{t-\tau} \mathbb{E}\left[ \left| \nabla_i F_\mu(\boldsymbol{\theta}_{\tau-1}) \right|^2 \right] .$$
$$\tag{20}$$

**Remark.** Thm. 5.4 introduces a novel variance-dependent upper bound for our refined second moment estimate (6). Compared with the bound (20) in Cor. 5.5 for the conventional estimate (5), our (6) reduces the influence of gradient estimation variance $\Sigma^2$ (in green) by a factor of $\frac{1-\beta_1}{1+\beta_1}$. This is crucial in ZO optimization where $\Sigma^2$ is typically large. While our estimate introduces a bias (in orange), it is small with a small learning rate $\eta$. Note that (18) represents the variance-free ideal, which the conventional estimate (5) and our refined estimate (6) aims to approximate. Comparing

the bounds in (19) and (20) with (18), our refined estimate (6) better approaches this ideal than (5), particularly when $\Sigma^2$ dominates, thanks to its reduced impact of $\Sigma^2$. This thus enables a better capture of geometry information during optimization and probably leads to improved optimization.

### 5.3. Variance-Aware Convergence Analysis

This section presents the first variance-aware convergence framework for adaptive ZO methods, particularly focusing on the convergence of $\mathcal{R}$-AdaZO and ZO-AdaMM. We first bound the averaged gradient norm of the smoothed function, $F_\mu$, as a step towards bounding the averaged gradient norm of the original function $F$. Inspired by (Zhang et al., 2024a), the core proof idea lies in applying Hölder's inequality to decomposes this target into two components (Lemma 5.6): One involving the averaged square root norm of second moment estimate that will be variance-dependent and another involving a normalized gradient norm by second moment estimate. The subsequent analysis then focuses on bounding these two components using Lemma 5.7 and Thm. 5.8, respectively. The first component, i.e., the averaged square root norm of second moment, can be upper-bounded with Thm. 5.4 (see Lemma 5.7). The second component, i.e., the normalized gradient norm with second moment, is shown to be of order $\mathcal{O}(\epsilon^2)$ (see Thm. 5.8). By combining these bounds and incorporating the connection between $\nabla F$ and $\nabla F_\mu$ in Lemma 5.1, we arrive at the final convergence results for $\mathcal{R}$-AdaZO (Thm. 5.9) and ZO-AdaMM (Thm. 5.10).

We first introduce Lemma 5.6 (proof in Appx. A.6).

**Lemma 5.6.** $\forall t \geq 1$, *we have that*

$$
\left( \frac{1}{T} \sum_{t=0}^{T-1} \mathbb{E}\left[\|\nabla F_\mu(\boldsymbol{\theta}_t)\|\right] \right)^2 \leq
$$
$$
\underbrace{\frac{1}{T} \sum_{t=0}^{T-1} \mathbb{E}\left[\sqrt{\beta_2 \|\boldsymbol{v}_t\| + \zeta}\right]}_{\textcircled{A}} \underbrace{\frac{1}{T} \sum_{t=0}^{T-1} \mathbb{E}\left[\frac{\|\nabla F_\mu(\boldsymbol{\theta}_t)\|^2}{\sqrt{\beta_2 \|\boldsymbol{v}_t\| + \zeta}}\right]}_{\textcircled{B}} .
$$
(21)

**Remark.** Chen et al. (2019); Nazari et al. (2020) bound $\textcircled{B}$ solely to demonstrate the convergence of adaptive ZO methods. However, we argue that this bound alone fail to include the effects of second moment estimate and therefore provides incomplete convergence information. In contrast, Lemma 5.6 allows us to include the effects of second moment (i.e., $\textcircled{A}$) and directly bound a more relevant quantity, $\frac{1}{T} \sum_{t=0}^{T-1} \mathbb{E}\left[\|\nabla F_\mu(\boldsymbol{\theta}_t)\|\right]$. Note that this metric is a more widely accepted convergence criteria in optimization theory, directly measuring the distance to a stationary point (Arjevani et al., 2023; Zhang et al., 2024a). Overall, Lemma 5.6 enables us to provide a variance-aware convergence analysis,

strengthening the understanding of convergence behavior for adaptive ZO methods.

Leveraging Lemma 5.6, we then proceed to bound the terms $\textcircled{A}$ and $\textcircled{B}$ in Lemma 5.7 (proof in Appx. A.7) and Lemma 5.8 (proof in Appx. A.8), respectively. These results rely on the following definition of $V$ resulted from Thm. 5.4.

$$
V^2 \triangleq \|\boldsymbol{v}_0\| + \underbrace{(1-\beta_1)\Sigma^2}_{\text{Variance}} + \underbrace{\frac{\beta_1(1+\beta_1)^2 L^2 \eta^2 d}{(1-\beta_1)^2(1-\beta_2)}}_{\text{Squared Bias}} . \quad (22)
$$

**Lemma 5.7.** *Given first and second moment estimates* (4) *and* (6) *respectively, with Assump.* 1, 2, $\forall t \geq 1, \forall i \in [d]$,

$$
\frac{1}{T} \sum_{t=0}^{T-1} \mathbb{E}\left[\sqrt{\beta_2 \|\boldsymbol{v}_t\| + \zeta}\right] \leq
$$
$$
\sqrt{\zeta} + Vd + \frac{(1+\beta_1)\sqrt{d}}{\sqrt{\beta_1(1-\beta_2)}} \frac{1}{T} \sum_{t=0}^{T-1} \mathbb{E}\left[\|\nabla F_\mu(\boldsymbol{\theta}_t)\|\right] .
$$
(23)

**Remark.** Lemma 5.7 demonstrates that $\textcircled{A}$ in Lemma 5.6 is variance-dependent. Specifically, $\textcircled{A}$ is asymptotically dominated by $V$ as $T \to \infty$, because $\frac{1}{T} \sum_{t=0}^{T-1} \mathbb{E}\left[\|\nabla F_\mu(\boldsymbol{\theta}_t)\|\right]$ gradually decreases during optimization. This highlights that the asymptotic behavior of $\textcircled{A}$ is governed by both the bias and variance present in the first moment estimate (4).

**Theorem 5.8** (Informal). *Given Assump.* 1, 2, *let* $1-\beta_2 \sim \mathcal{O}(\epsilon^2)$, $\eta \sim \mathcal{O}(\epsilon^2)$, *and* $T \sim \mathcal{O}(\epsilon^{-4})$. *the following holds for $\mathcal{R}$-AdaZO if* $\beta_1 \leq \sqrt{\beta_2}, \beta_2 \geq \frac{1}{2}, \boldsymbol{m}_{0,i} = 0, \boldsymbol{v}_{0,i} > 0$,

$$
\frac{1}{T} \sum_{t=0}^{T-1} \mathbb{E}\left[\frac{\|\nabla F_\mu(\boldsymbol{\theta}_{t-1})\|^2}{\sqrt{\beta_2 \|\boldsymbol{v}_t\| + \zeta}}\right] \leq \epsilon^2 . \quad (24)
$$

**Remark.** Of note, Thm. 5.8 attains the same rate of $\mathcal{O}(\frac{1}{\sqrt{T}})$ as (Chen et al., 2019; Nazari et al., 2020) to achieve that $\frac{1}{T} \sum_{t=0}^{T-1} \mathbb{E}\left[\frac{\|\nabla F_\mu(\boldsymbol{\theta}_{t-1})\|^2}{\sqrt{\beta_2 \|\boldsymbol{v}_t\| + \zeta}}\right] \leq \epsilon$.

By incorporating Lemma 5.1, Lemma 5.7, and Thm. 5.8 into Lemma 5.6, we derive the following convergence for $\mathcal{R}$-AdaZO (Thm. 5.9, proof in Appx. A.9) and ZO-AdaMM (Thm. 5.10, proof in Appx. A.10), respectively.

**Theorem 5.9** (Informal). *Given Assump.* 1, 2, *let* $1-\beta_2 \sim \mathcal{O}(\epsilon^2)$, $\eta \sim \mathcal{O}(\epsilon^2)$, *and* $T \sim \mathcal{O}(\epsilon^{-4})$. *We have the following for $\mathcal{R}$-AdaZO if* $\beta_1 \leq \sqrt{\beta_2}, \beta_2 \geq 1/2$ *and* $\forall i \in [d], \boldsymbol{m}_{0,i} = 0, \boldsymbol{v}_{0,i} > 0$,

$$
\frac{1}{T} \sum_{t=0}^{T-1} \mathbb{E}\left[\|\nabla F(\boldsymbol{\theta}_t)\|\right] \leq \frac{(1+\beta_1)\sqrt{d}}{\sqrt{\beta_1(1-\beta_2)}} \epsilon^2 + \left(\sqrt[4]{\zeta} + \sqrt{Vd}\right)\epsilon
$$
$$
+ \mu L \sqrt{d}
$$
(25)

*where $V$ is defined in* (22).

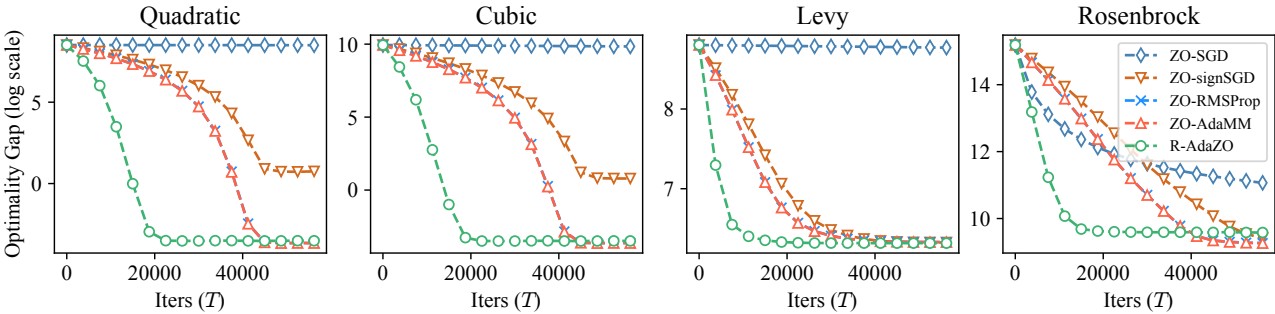

Figure 1: Convergence comparison among different adaptive ZO optimizers for various synthetic functions, in which $y$-axis represents the log-scale optimality gap $F(\boldsymbol{\theta}) - \min_{\boldsymbol{\theta}'} F(\boldsymbol{\theta}')$ and $x$-axis is the number of iterations $T$. Each curve denotes the mean from 3 independent runs.

**Theorem 5.10** (Informal). *Given Assump. 1, 2, let $1 - \beta_2 \sim \mathcal{O}(\epsilon^2)$, $\eta \sim \mathcal{O}(\epsilon^2)$, and $T \sim \mathcal{O}(\epsilon^{-4})$. We have the following for ZO-AdaMM if $\beta_1 \leq \sqrt{\beta_2}, \beta_2 \geq 1/2$ and $\forall i \in [d], \boldsymbol{m}_{0,i} = 0, \boldsymbol{v}_{0,i} > 0$,*

$$\frac{1}{T}\sum_{t=0}^{T-1}\mathbb{E}\big[\,\|\nabla F(\boldsymbol{\theta}_t)\|\,\big] \leq \frac{(1+\beta_1)\sqrt{d}}{\sqrt{\beta_1(1-\beta_2)}}\epsilon^2 + \left(\sqrt[4]{\zeta}+\sqrt{\hat{V}d}\right)\epsilon + \mu L\sqrt{d} \tag{26}$$

*where $\hat{V}^2 \triangleq \|\boldsymbol{v}_0\| + \underbrace{(1+\beta_1)\Sigma^2}_{\text{Variance}}$.*

**Remark.** To the best of our knowledge, our Thm. 5.9 and Thm. 5.10 are the first analyses to explicitly incorporate the impact of second moment estimate (measured by $V$ or $\hat{V}$) that is variance-dependent into the convergence of adaptive ZO methods. Specifically, Thm. 5.9 and Thm. 5.10 demonstrate that the convergence of $\frac{1}{T}\sum_{t=0}^{T-1}\mathbb{E}\big[\,\|\nabla F(\boldsymbol{\theta}_t)\|\,\big]$ typically exhibits a dependence of $\mathcal{O}(\sqrt{V}\epsilon)$ in adaptive ZO methods, highlighting the importance of an improved second moment estimate with reduced variance. This explains the advantage of $\mathcal{R}$-AdaZO over other adaptive ZO methods like ZO-AdaMM thanks to our refined second moment estimate (6) achieving a reduction of at most $\frac{1-\beta_1}{1+\beta_1}$ on $V$. Moreover, Thm. 5.9 implies that a larger $\beta_1$ typically leads to an improved convergence of $\mathcal{R}$-AdaZO especially when variance $\Sigma^2$ dominates. Comparing Thm. 5.9 and Cor. 5.10, we observe that $\mathcal{R}$-AdaZO achieves a speedup of $\mathcal{O}\left(\sqrt[4]{\frac{1+\beta_1}{1-\beta_1}}\right)$ for the convergence of averaged gradient norm primarily when variance $\Sigma^2$ dominates. These will be further empirically supported by the experiments in our Sec. 6 below.

## 6. Experiments

In this section, we conduct extensive experiments on various tasks in practice, including synthetic functions (Sec. 6.1),

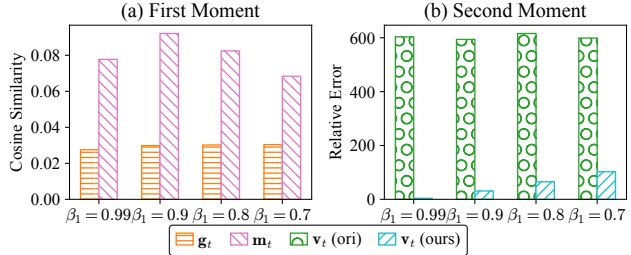

Figure 2: Effects of (a) first and (b) second moment under varying $\beta_1$ during the convergence of Quadratic function. Here, $\boldsymbol{g}_t$ and $\boldsymbol{m}_t$ corresponds to the results from the estimated gradient in (2) and the first moment in (4), and $\boldsymbol{v}_t$ (ori) and $\boldsymbol{v}_t$ (ours) are results of the second moment estimates defined in (5) and (6) respectively. The $y$-axis in (a) represents the cosine similarity between $\boldsymbol{g}_t$ or $\boldsymbol{m}_t$ and the true gradient $\nabla F(\boldsymbol{\theta}_{t-1})$, while the $y$-axis in (b) denotes the relative error between $\boldsymbol{v}_t$ in (5) or (6) and the $\boldsymbol{v}_t$ computed using the square of the true gradient $\nabla F(\boldsymbol{\theta}_{t-1})$.

black-box adversarial attack (Sec. 6.2), as well as memory-efficient fine-tuning of large language models (Sec. 6.3), to empirically support the superior efficacy of our $\mathcal{R}$-AdaZO (Algo. 2).

### 6.1. Synthetic Functions

**On Convergence.** We first compare the convergence of $\mathcal{R}$-AdaZO with ZO-SGD, ZO-signSGD (Liu et al., 2019), ZO-AdaMM, and ZO-RMSProp, an integration of RMSProp (Hinton, 2012) and ZO gradient estimator, using four synthetic functions with $d=10^4$, including Quadratic, Cubic, Levy, and Rosenbrock function. We refer to Appx. B.1 for more details. The results are in Fig. 1, showing that $\mathcal{R}$-AdaZO generally achieves significantly faster convergence while maintaining competitive optimality gaps compared to other baselines. The consistent gain of $\mathcal{R}$-AdaZO across most synthetic functions suggests that $\mathcal{R}$-AdaZO is robust

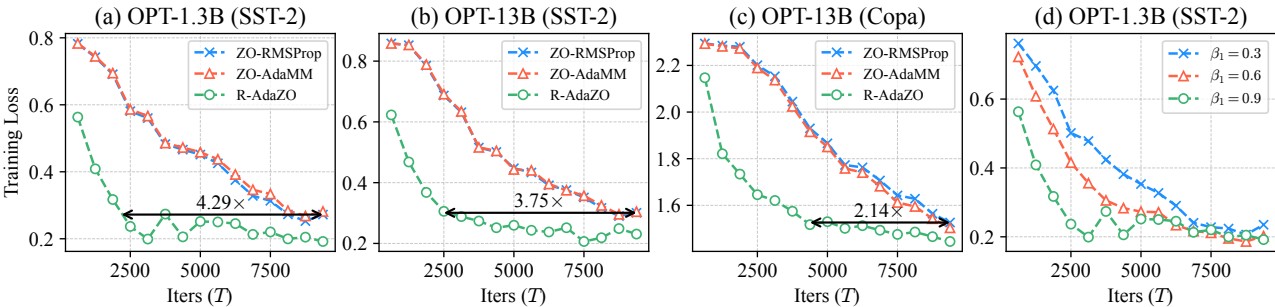

Figure 3: (a-c) Training loss comparison among various adaptive ZO optimizers for the fine-tuning of LLMs under different model sizes and different dataset. (d) Training loss comparison of varying $\beta_1$ in $\mathcal{R}$-AdaZO on OPT-1.3B and SST-2 (Socher et al., 2013). Each curve denotes the mean from 3 independent runs.

to the structure of the underlying problem. Furthermore, Fig. 1 reveals a notable similarity in the convergence behavior of ZO-AdaMM and ZO-RMSProp across all four benchmark functions. In contrast, $\mathcal{R}$-AdaZO consistently demonstrates a substantial speedup compared to ZO-RMSProp. These results imply that the first moment itself contributes minimally to the convergence gains for adaptive ZO optimization, and underscores the critical role of our refined second moment estimate in achieving the superior performance of $\mathcal{R}$-AdaZO.

**On First Moment.** We further conduct an experimental analysis to understand how $\beta_1$ affects first moment estimates during the optimization process of the Quadratic function. In Fig. 2 (a), we present the results, using cosine similarity to measure the alignment between the estimated gradient $\boldsymbol{g}_t$ or the estimated first moment $\boldsymbol{m}_t$, and the true gradient $\nabla F(\boldsymbol{\theta}_{t-1})$. The results indicate that the estimated first moment $\boldsymbol{m}_t$ exhibits better cosine similarity than $\boldsymbol{g}_t$, resulting from its variance reduction effect, as proven in Thm. 5.3. Moreover, we observe that increasing $\beta_1$ generally enhances this variance reduction. However, excessively high values of $\beta_1$ result in a minor decrease in similarity. This trend is consistent with the trade-off discussed in Thm. 5.3.

**On Second Moment.** We further conduct an experimental analysis to understand how $\beta_1$ affects second moment estimates during the optimization process of the Quadratic function. Figure 2(b) compares the second moment estimates, $\boldsymbol{v}_t$(ori) from (5) and $\boldsymbol{v}_t$(ours) from (6), using the relative error against the second moment estimate based on the squared ground truth $(\nabla F(\boldsymbol{\theta}_{t-1}))^2$. The results demonstrate that our refined second moment estimate, $\boldsymbol{v}_t$(ours), significantly reduces the relative error compared to the standard second moment estimate, $\boldsymbol{v}_t$(ori), which therefore enables the capture of more accurate geometry information during optimization. Interestingly, increasing values of $\beta_1$ generally lead to a lower relative error, a trend that contrasts with the behavior of first moment estimates. This lack of a

Table 1: Comparison of the number of iterations to achieve a successful black-box adversarial attack. Each cell represents mean $\pm$ standard deviation from five independent runs.

| Measurement | ZO-RMSProp | ZO-AdaMM | $\mathcal{R}$-AdaZO |
|---|---|---|---|
| # Iters ($\times 10^3$) | 15.6$\pm$3.2 | 15.5$\pm$4.1 | **2.9$\pm$0.8** |
| Speedup | 1.0$\times$ | 1.0$\times$ | **5.4$\times$** |

trade-off is likely due to the loose bound we derived for our refined second moment.

### 6.2. Black-Box Adversarial Attack

We further investigate the performance of $\mathcal{R}$-AdaZO within the realm of black-box adversarial attacks, a key application area for zeroth-order optimization (Cheng et al., 2021; Shu et al., 2023). In this context, the objective is to find an optimal perturbation $\delta$ that causes a black-box model to misclassify an input image $\mathbf{x}$ when perturbed to $\mathbf{x} + \delta$. Following the practice in (Shu et al., 2023), we also present a comparative analysis of the number of iterations required for successful black-box adversarial attacks on an image from the MNIST dataset (Lecun et al., 1998), using ZO-RMSProp, ZO-AdaMM, and $\mathcal{R}$-AdaZO in Tab. 1 (experimental setup in Appx. B.2). As shown in the table, ZO-RMSProp and ZO-AdaMM exhibit similar performance, requiring an average of approximately 15.6 and 15.5 thousand iterations, respectively. The standard deviations of the iteration counts were similar as well, about 3200 to 4100 iterations. These align with our results in Sec. 6.1. On the other hand, $\mathcal{R}$-AdaZO requires a significantly lower number of iterations with an average of only 2900, and a smaller standard deviation of 800 iterations, suggesting a faster and more stable convergence. The speedup achieved by $\mathcal{R}$-AdaZO, i.e., a speedup of 5.4$\times$ compared to the baseline ZO-RMSProp, is also highlighted in Tab. 1. These findings thus further underscore the superior efficacy of $\mathcal{R}$-AdaZO.

## 6.3. Memory-Efficient LLM Fine-Tuning

Recent interest in memory-efficient fine-tuning of large language models using ZO optimization (Malladi et al., 2023; Zhang et al., 2024b) motivates our use of this setting to further demonstrate the superiority of $\mathcal{R}$-AdaZO over other adaptive ZO optimization algorithms (experimental setup in Appx. B.3). The results in Fig. 3(a-c) show that, for both OPT-1.3B and OPT-13B models (Zhang et al., 2022) and dataset SST-2 (Socher et al., 2013) and Copa (Roemmele et al., 2011), $\mathcal{R}$-AdaZO converges significantly faster than ZO-RMSProp and ZO-AdaMM on SST-2 dataset, achieving a speedup of $4.29\times$ for OPT-1.3B and $3.75\times$ for OPT-13B to reach the same training loss. The optimization curves of ZO-RMSProp and ZO-AdaMM are indistinguishable, indicating the similar convergence behavior we have seen in Sec. 6.1 and Sec. 6.2. These empirical results strongly support $\mathcal{R}$-AdaZO as a more efficient and effective adaptive ZO optimizer. In addition, the results in Fig. 3(d) show that $\mathcal{R}$-AdaZO typically enjoys an improved convergence when a larger $\beta_1$ is applied, which aligns with the insights provided by our Thm. 5.9.

## 7. Conclusion

In conclusion, this work introduces $\mathcal{R}$-AdaZO, a novel approach that addresses the critical limitations of existing adaptive ZO methods by effectively leveraging moment information. Through rigorous theoretical analysis, we have demonstrated the inherent variance reduction effect of first moment estimates on ZO gradient estimates, leading to more stable and accurate updates, as well as the improved accuracy of our refined second moment estimates. Furthermore, we establish the first variance-aware convergence framework for adaptive ZO methods and prove the superior convergence rate of $\mathcal{R}$-AdaZO. The consistent empirical performance gains of $\mathcal{R}$-AdaZO across diverse applications underscore its potential as a powerful and practical solution for real-world ZO optimization challenges.

## Impact Statement

This paper presents work whose goal is to advance the field of zeroth-order optimization and machine learning. There are many potential societal consequences of our work, none of which we feel must be specifically highlighted here.

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

## A. Proofs

### A.1. Proof of Lemma 5.1

Based on the definition of $\hat{\nabla} f(\boldsymbol{\theta}, \xi)$ in (2), we first prove (12) in Lemma 5.1 as below,

$$
\begin{aligned}
\mathbb{E}\left[\hat{\nabla} f(\boldsymbol{\theta}, \xi)\right] &= \frac{d}{K} \sum_{k=1}^{K} \mathbb{E}_{\boldsymbol{u}_k \sim \mathbb{S}^{d-1}}\left[\mathbb{E}_{\xi}\left[\frac{f(\boldsymbol{\theta} + \mu \boldsymbol{u}_k; \xi) - f(\boldsymbol{\theta}; \xi)}{\mu} \boldsymbol{u}_k\right]\right] \\
&= \frac{d}{K} \sum_{k=1}^{K}\left(\mathbb{E}_{\boldsymbol{u}_k \sim \mathbb{S}^{d-1}}\left[\frac{F(\boldsymbol{\theta} + \mu \boldsymbol{u}_k)}{\mu} \boldsymbol{u}_k\right] - \mathbb{E}_{\boldsymbol{u}_k \sim \mathbb{S}^{d-1}}\left[\frac{F(\boldsymbol{\theta})}{\mu} \boldsymbol{u}_k\right]\right) \\
&= \frac{1}{K} \sum_{k=1}^{K} \nabla F_\mu(\boldsymbol{\theta}) \\
&= \nabla F_\mu(\boldsymbol{\theta})
\end{aligned}
\tag{27}
$$

where the third equality is due to the fact that $\mathbb{E}_{\boldsymbol{u}_k \sim \mathbb{S}^{d-1}}\left[\frac{F(\boldsymbol{\theta} + \mu \boldsymbol{u}_k)}{\mu} \boldsymbol{u}_k\right] = \frac{\nabla F_\mu(\boldsymbol{\theta})}{d}$, which comes from Lemma 1 in (Flaxman et al., 2004) with $F_\mu$ defined on a unit ball $\mathbb{B}^d$ as shown in (11).

We then prove (13) in Lemma 5.1 as below,

$$
\begin{aligned}
\mathbb{E}\left[\left\|\nabla F(\boldsymbol{\theta}) - \nabla F_\mu(\boldsymbol{\theta})\right\|\right] &\overset{(a)}{=} \mathbb{E}\left[\left\|\mathbb{E}_{\boldsymbol{u} \sim \mathbb{B}^d}\left[\nabla F(\boldsymbol{\theta}) - \nabla F(\boldsymbol{\theta} + \mu \boldsymbol{u})\right]\right\|\right] \\
&\overset{(b)}{\leq} \mathbb{E}\left[\left\|\nabla F(\boldsymbol{\theta}) - \nabla F(\boldsymbol{\theta} + \mu \boldsymbol{u})\right\|\right] \\
&\overset{(c)}{\leq} \mathbb{E}\left[\mu L \sqrt{d}\left\|\boldsymbol{u}\right\|\right] \\
&\overset{(d)}{=} \mu L \sqrt{d}
\end{aligned}
\tag{28}
$$

where $(a)$ comes from the Leibniz's Rule, $(b)$ results from Jensen's inequality, $(c)$ is based on Assump. 1, and $(d)$ is due to the fact that $\|\boldsymbol{u}\| \leq 1$. We therefore conclude our proof for Lemma 5.1.

### A.2. Proof of Lemma 5.2

With Leibniz's Rule, Jensen's inequality, and $(d)$ Assump. 1, the following holds for (14) in Lemma 5.2:

$$
\begin{aligned}
\left|\nabla_i F_\mu(\boldsymbol{\theta}) - \nabla_i F_\mu(\boldsymbol{\theta}')\right| &= \left|\nabla_i \mathbb{E}_{\boldsymbol{u} \sim \mathbb{B}^d}\left[F(\boldsymbol{\theta} + \mu \boldsymbol{u}) - F(\boldsymbol{\theta}' + \mu \boldsymbol{u})\right]\right| \\
&= \left|\mathbb{E}_{\boldsymbol{u} \sim \mathbb{B}^d}\left[\nabla_i F(\boldsymbol{\theta} + \mu \boldsymbol{u}) - \nabla_i F(\boldsymbol{\theta}' + \mu \boldsymbol{u})\right]\right| \\
&\leq \mathbb{E}_{\boldsymbol{u} \sim \mathbb{B}^d}\left[\left|\nabla_i F(\boldsymbol{\theta} + \mu \boldsymbol{u}) - \nabla_i F(\boldsymbol{\theta}' + \mu \boldsymbol{u})\right|\right] \\
&\leq L\left\|\boldsymbol{\theta} - \boldsymbol{\theta}'\right\| .
\end{aligned}
\tag{29}
$$

We finally prove (15) in Lemma 5.2 as below,

$$
\mathbb{E}\left[\left|\hat{\nabla}_i f(\boldsymbol{\theta}, \xi) - \nabla_i F_\mu(\boldsymbol{\theta})\right|^2\right]
$$

$$
\stackrel{(a)}{=} \frac{d^2}{K^2}\mathbb{E}\left[\left(\sum_{k=1}^{K}\left(\frac{f(\boldsymbol{\theta}+\mu\boldsymbol{u}_k,\xi)}{\mu}\boldsymbol{u}_{k,i} - \mathbb{E}_{\boldsymbol{u}_k\sim\mathbb{S}^{d-1}}\left[\frac{F(\boldsymbol{\theta}+\mu\boldsymbol{u}_k)}{\mu}\boldsymbol{u}_{k,i}\right]\right) - \left(\frac{f(\boldsymbol{\theta},\xi)}{\mu}\boldsymbol{u}_{k,i} - \mathbb{E}_{\boldsymbol{u}_k\sim\mathbb{S}^{d-1}}\left[\frac{F(\boldsymbol{\theta})}{\mu}\boldsymbol{u}_{k,i}\right]\right)\right)^2\right]
$$

$$
\stackrel{(b)}{=} \frac{d^2}{K^2}\sum_{k=1}^{K}\mathbb{E}\left[\left(\left(\frac{f(\boldsymbol{\theta}+\mu\boldsymbol{u}_k,\xi)}{\mu}\boldsymbol{u}_{k,i} - \mathbb{E}_{\boldsymbol{u}_k\sim\mathbb{S}^{d-1}}\left[\frac{F(\boldsymbol{\theta}+\mu\boldsymbol{u}_k)}{\mu}\boldsymbol{u}_{k,i}\right]\right) - \left(\frac{f(\boldsymbol{\theta},\xi)}{\mu}\boldsymbol{u}_{k,i} - \mathbb{E}_{\boldsymbol{u}_k\sim\mathbb{S}^{d-1}}\left[\frac{F(\boldsymbol{\theta})}{\mu}\boldsymbol{u}_{k,i}\right]\right)\right)^2\right]
$$

$$
\stackrel{(c)}{\leq} \frac{2d^2}{K^2}\sum_{k=1}^{K}\mathbb{E}\left[\left(\frac{f(\boldsymbol{\theta}+\mu\boldsymbol{u}_k,\xi)-F(\boldsymbol{\theta}+\mu\boldsymbol{u}_k)}{\mu}\boldsymbol{u}_{k,i} + \frac{F(\boldsymbol{\theta}+\mu\boldsymbol{u}_k)}{\mu}\boldsymbol{u}_{k,i} - \mathbb{E}_{\boldsymbol{u}_k\sim\mathbb{S}^{d-1}}\left[\frac{F(\boldsymbol{\theta}+\mu\boldsymbol{u}_k)}{\mu}\boldsymbol{u}_{k,i}\right]\right)^2\right] +
$$
$$
\mathbb{E}\left[\left(\frac{f(\boldsymbol{\theta},\xi)-F(\boldsymbol{\theta})}{\mu}\boldsymbol{u}_{k,i} + \frac{F(\boldsymbol{\theta})}{\mu}\boldsymbol{u}_{k,i} - \mathbb{E}_{\boldsymbol{u}_k\sim\mathbb{S}^{d-1}}\left[\frac{F(\boldsymbol{\theta})}{\mu}\boldsymbol{u}_{k,i}\right]\right)^2\right]
$$

$$
\stackrel{(d)}{=} \frac{4d^2}{K^2}\sum_{k=1}^{K}\mathbb{E}\left[\left(\frac{f(\boldsymbol{\theta}+\mu\boldsymbol{u}_k,\xi)-F(\boldsymbol{\theta}+\mu\boldsymbol{u}_k)}{\mu}\boldsymbol{u}_{k,i}\right)^2 + \left(\frac{F(\boldsymbol{\theta}+\mu\boldsymbol{u}_k)}{\mu}\boldsymbol{u}_{k,i} - \mathbb{E}_{\boldsymbol{u}_k\sim\mathbb{S}^{d-1}}\left[\frac{F(\boldsymbol{\theta}+\mu\boldsymbol{u}_k)}{\mu}\boldsymbol{u}_{k,i}\right]\right)^2\right] +
$$
$$
\mathbb{E}\left[\left(\frac{f(\boldsymbol{\theta},\xi)-F(\boldsymbol{\theta})}{\mu}\boldsymbol{u}_{k,i}\right)^2 + \left(\frac{F(\boldsymbol{\theta})}{\mu}\boldsymbol{u}_{k,i} - \mathbb{E}_{\boldsymbol{u}_k\sim\mathbb{S}^{d-1}}\left[\frac{F(\boldsymbol{\theta})}{\mu}\boldsymbol{u}_{k,i}\right]\right)^2\right]
$$

$$
\stackrel{(e)}{\leq} \frac{4d^2}{K^2}\sum_{k=1}^{K}\mathbb{E}\left[\frac{\sigma^2}{\mu^2}\boldsymbol{u}_{k,i}^2 + \left(\frac{F(\boldsymbol{\theta}+\mu\boldsymbol{u}_k)}{\mu}\boldsymbol{u}_{k,i}\right)^2 + \frac{\Sigma^2}{\mu^2}\boldsymbol{u}_{k,i}^2 + \left(\frac{F(\boldsymbol{\theta})}{\mu}\boldsymbol{u}_{k,i}\right)^2\right]
$$

$$
\stackrel{(f)}{\leq} \frac{8(\sigma^2+C^2)d}{\mu^2 K}.
$$

$$(30)$$

Here, $(a)$ is due to the fact that $\mathbb{E}_{\boldsymbol{u}_k\sim\mathbb{S}^{d-1}}\left[\frac{F(\boldsymbol{\theta}+\mu\boldsymbol{u}_k)}{\mu}\boldsymbol{u}_k\right] = \frac{\nabla F_\mu(\boldsymbol{\theta})}{d}$, which comes from Lemma 1 in (Flaxman et al., 2004) with $F_\mu$ defined on a unit ball $\mathbb{B}^d$ as shown in (11), and the fact that $\mathbb{E}_{\boldsymbol{u}_k\sim\mathbb{S}^{d-1}}\left[\frac{F(\boldsymbol{\theta})}{\mu}\boldsymbol{u}_{k,i}\right] = \boldsymbol{0}$. In addition, $(b)$ comes from the independence among $\{\boldsymbol{u}_k\}_{k=1}^{K}$, and $(c),(d)$ are from Cauchy-Schwarz inequality. Besides, $(e)$ results from Assump. 2 and the definition of variance. Finally, $(f)$ is due to Assump. 1 and the fact that $\mathbb{E}\left[\boldsymbol{u}_{k,i}^2\right] = 1/d$. We therefore conclude our proof for Lemma 5.2.

### A.3. Proof of Thm. 5.3

We first show the following variance reduction effect in first moment estimate based on the definition of $\Sigma^2$ in (16):

$$
\mathbb{E}\left[\left|\boldsymbol{m}_{t,i}-\mathbb{E}[\boldsymbol{m}_{t,i}]\right|^2\right] \stackrel{(a)}{=} \mathbb{E}\left[\left|(1-\beta_1)\sum_{\tau=1}^{t}\beta_1^{t-\tau}\left(\hat{\nabla}_i f(\boldsymbol{\theta}_{\tau-1},\xi_\tau) - \nabla_i F_\mu(\boldsymbol{\theta}_{\tau-1})\right)\right|^2\right]
$$
$$
\stackrel{(b)}{=} (1-\beta_1)^2\sum_{\tau=1}^{t}\beta_1^{2(t-\tau)}\mathbb{E}\left[\left|\hat{\nabla}_i f(\boldsymbol{\theta}_{\tau-1},\xi_\tau) - \nabla_i F_\mu(\boldsymbol{\theta}_{\tau-1})\right|^2\right]
$$
$$
\stackrel{(c)}{\leq} \frac{(1-\beta_1)(1-\beta_1^{2t})}{1+\beta_1}\Sigma^2
$$
$$
\stackrel{(d)}{\leq} \frac{1-\beta_1}{1+\beta_1}\Sigma^2
$$

$$(31)$$

where $(b)$ comes from the independence among $\{\xi_\tau\}_{\tau=1}^{t}$ and $(c)$ results from Lemma 5.2.

**Remark.** As suggested by (31), the standard bias correction term (i.e., $1-\beta_1^t$) in Adam (Kingma & Ba, 2015) is intentionally excluded to avoid compromising the variance reduction effect.

We then show the bias in the first moment estimate as below,

$$\mathbb{E}\left[\left\|\frac{1}{1-\beta_1^t}\mathbb{E}[\boldsymbol{m}_{t,i}] - \nabla_i F_\mu(\boldsymbol{\theta}_{t-1})\right\|^2\right]$$

$$\overset{(a)}{=} \mathbb{E}\left[\left\|\frac{(1-\beta_1)}{1-\beta_1^t}\sum_{\tau=1}^{t}\beta_1^{t-\tau}\left(\nabla_i F_\mu(\boldsymbol{\theta}_{\tau-1}) - \nabla_i F_\mu(\boldsymbol{\theta}_{t-1})\right)\right\|^2\right]$$

$$\overset{(b)}{=} \left(\frac{1-\beta_1}{1-\beta_1^t}\right)^2 \sum_{\tau,\tau'=1}^{t} \mathbb{E}\left[\left\langle\beta_1^{t-\tau}(\nabla_i F_\mu(\boldsymbol{\theta}_{\tau-1}) - \nabla_i F_\mu(\boldsymbol{\theta}_{t-1})), \beta_1^{t-\tau'}(\nabla_i F_\mu(\boldsymbol{\theta}_{\tau'-1}) - \nabla_i F_\mu(\boldsymbol{\theta}_{t-1}))\right\rangle\right]$$

$$\overset{(c)}{\leq} \left(\frac{1-\beta_1}{1-\beta_1^t}\right)^2 \sum_{\tau,\tau'=1}^{t} \frac{\beta_1^{2t-\tau-\tau'}}{2}\left(\mathbb{E}\left[\left|\nabla_i F_\mu(\boldsymbol{\theta}_{\tau-1}) - \nabla_i F_\mu(\boldsymbol{\theta}_{t-1}))\right|^2\right] + \mathbb{E}\left[\left|\nabla_i F_\mu(\boldsymbol{\theta}_{\tau'-1}) - \nabla_i F_\mu(\boldsymbol{\theta}_{t-1}))\right|^2\right]\right)$$

$$\overset{(d)}{=} \left(\frac{1-\beta_1}{1-\beta_1^t}\right)^2 \sum_{\tau=1}^{t} \frac{\beta_1^{t-\tau}(1-\beta_1^t)}{1-\beta_1}\mathbb{E}\left[\left|\nabla_i F_\mu(\boldsymbol{\theta}_{\tau-1}) - \nabla_i F_\mu(\boldsymbol{\theta}_{t-1}))\right|^2\right] \tag{32}$$

$$\overset{(e)}{\leq} \frac{(1-\beta_1)L^2}{1-\beta_1^t}\sum_{\tau=1}^{t-1}\beta_1^{t-\tau}\mathbb{E}\left[\left\|\boldsymbol{\theta}_{\tau-1} - \boldsymbol{\theta}_{t-1}\right\|^2\right]$$

$$\overset{(f)}{\leq} \frac{(1-\beta_1)L^2\eta^2}{1-\beta_1^t}\sum_{\tau=1}^{t-1}\beta_1^{t-\tau}(t-\tau)\sum_{i=1}^{d}\sum_{s=\tau}^{t-1}\mathbb{E}\left[\frac{\boldsymbol{m}_{s,i}^2}{\boldsymbol{v}_{s,i}+\zeta}\right]$$

$$\overset{(g)}{\leq} \frac{(1-\beta_1)L^2\eta^2 d}{(1-\beta_1^t)(1-\beta_2)}\sum_{\tau=1}^{t-1}\beta_1^{t-\tau}(t-\tau)^2$$

$$\overset{(h)}{\leq} \frac{\beta_1(1+\beta_1)L^2\eta^2 d}{(1-\beta_1^t)(1-\beta_1)^2(1-\beta_2)}$$

where $(c)$ is from Cauchy-Schwarz inequality, $(d)$ is from the sum of geometric series, $(e)$ is from (14) in Lemma 5.2, $(f)$ is based on the update rule in (7) and Cauchy-Schwarz inequality, $(g)$ is due to the fact that $\frac{\boldsymbol{m}_{s,i}^2}{\boldsymbol{v}_{s,i}+\zeta} \leq \frac{\boldsymbol{m}_{s,i}^2}{(1-\beta_2)\boldsymbol{m}_{s,i}^2}$. Finally, $(h)$ results from the following:

$$\sum_{\tau=1}^{t}\tau^2\beta_1^\tau = \frac{\beta_1\left(1+\beta_1 - (t+1)^2\beta_1^t + (2t^2+2t-1)\beta_1^{t+1} - t^2\beta_1^{t+2}\right)}{(1-\beta_1)^3} \leq \frac{\beta_1(1+\beta_1)}{(1-\beta_1)^3}. \tag{33}$$

By putting the results above together, we then conclude our proof for Thm. 5.3 as below:

$$\mathbb{E}\left[\left|\boldsymbol{m}_{t,i} - \nabla_i F_\mu(\boldsymbol{\theta}_{t-1})\right|^2\right]$$

$$\overset{(a)}{=} \mathbb{E}\left[\left|\boldsymbol{m}_{t,i} - \mathbb{E}\left[\boldsymbol{m}_{t,i}\right] + \mathbb{E}\left[\boldsymbol{m}_{t,i}\right] - \nabla_i F_\mu(\boldsymbol{\theta}_{t-1})\right|^2\right]$$

$$\overset{(b)}{=} \mathbb{E}\left[\left|\boldsymbol{m}_{t,i} - \mathbb{E}\left[\boldsymbol{m}_{t,i}\right]\right|^2\right] + \mathbb{E}\left[\left|\mathbb{E}\left[\boldsymbol{m}_{t,i}\right] - \nabla_i F_\mu(\boldsymbol{\theta}_{t-1})\right|^2\right] \tag{34}$$

$$\overset{(c)}{\leq} \mathbb{E}\left[\left|\boldsymbol{m}_{t,i} - \mathbb{E}\left[\boldsymbol{m}_{t,i}\right]\right|^2\right] + \left(1-\beta_1^t\right)\mathbb{E}\left[\left\|\frac{1}{1-\beta_1^t}\mathbb{E}[\boldsymbol{m}_{t,i}] - \nabla_i F_\mu(\boldsymbol{\theta}_{t-1})\right\|^2\right] + \beta_1^t\mathbb{E}\left[\left|\nabla_i F_\mu(\boldsymbol{\theta}_{t-1})\right|^2\right]$$

$$\overset{(d)}{\leq} \frac{1-\beta_1}{1+\beta_1}\Sigma^2 + \frac{\beta_1(1+\beta_1)L^2\eta^2 d}{(1-\beta_1)^2(1-\beta_2)} + \beta_1^t\mathbb{E}\left[\left|\nabla_i F_\mu(\boldsymbol{\theta}_{t-1})\right|^2\right]$$

where $(b)$ comes from the independence between $\boldsymbol{m}_{t,i} - \mathbb{E}\left[\boldsymbol{m}_{t,i}\right]$ and $\mathbb{E}\left[\boldsymbol{m}_{t,i}\right] - \nabla_i F_\mu(\boldsymbol{\theta}_{t-1})$ with respect to $\{\xi_\tau\}_\tau^t$, $(c)$ is due the fact that $(a+b)^2 \leq \left(1+\frac{1-\beta_1^t}{\beta_1^t}\right)a^2 + \left(1+\frac{\beta_1^t}{1-\beta_1^t}\right)b^2$.

### A.4. Proof of Thm. 5.4

Based on our Thm. 5.3, we naturally can bound $m_{t,i}^2$ as below

$$
\begin{aligned}
\mathbb{E}\left[\left|\boldsymbol{m}_{t,i}\right|^2\right] &= \mathbb{E}\left[\left|\boldsymbol{m}_{t,i} - \nabla_i F_\mu(\boldsymbol{\theta}_{t-1}) + \nabla_i F_\mu(\boldsymbol{\theta}_{t-1})\right|^2\right] \\
&\leq (1+\beta_1)\mathbb{E}\left[\left|\boldsymbol{m}_{t,i} - \nabla_i F_\mu(\boldsymbol{\theta}_{t-1})\right|^2\right] + \left(1+\frac{1}{\beta_1}\right)\mathbb{E}\left[\left|\nabla_i F_\mu(\boldsymbol{\theta}_{t-1})\right|^2\right] \\
&\leq (1-\beta_1)\Sigma^2 + \frac{\beta_1(1+\beta_1)^2 L^2 \eta^2 d}{(1-\beta_1)^2(1-\beta_2)} + \frac{(1+\beta_1)^2}{\beta_1}\mathbb{E}\left[\left|\nabla_i F_\mu(\boldsymbol{\theta}_{t-1})\right|^2\right] .
\end{aligned}
\tag{35}
$$

As $\boldsymbol{v}_{t,i}$ is the moving average of $\boldsymbol{m}_{t,i}$, we conclude our proof for Thm. 5.4 as below

$$
\begin{aligned}
\mathbb{E}\left[\boldsymbol{v}_{t,i}\right] &= \mathbb{E}\left[\beta_2 \boldsymbol{v}_{t-1,i} + (1-\beta_2)\boldsymbol{m}_{t,i}^2\right] \\
&\leq \mathbb{E}[\beta_2 \boldsymbol{v}_{t-1,i}] + (1-\beta_2)\left((1-\beta_1)\Sigma^2 + \frac{\beta_1(1+\beta_1)^2 L^2 \eta^2 d}{(1-\beta_1)^2(1-\beta_2)} + \frac{(1+\beta_1)^2}{\beta_1}\mathbb{E}\left[\left|\nabla_i F_\mu(\boldsymbol{\theta}_{t-1})\right|^2\right]\right) \\
&\leq \beta_2^t \boldsymbol{v}_{0,i} + \sum_{\tau=1}^{t}(1-\beta_2)\beta_2^{t-\tau}\left((1-\beta_1)\Sigma^2 + \frac{\beta_1(1+\beta_1)^2 L^2 \eta^2 d}{(1-\beta_1)^2(1-\beta_2)} + \frac{(1+\beta_1)^2}{\beta_1}\mathbb{E}\left[\left|\nabla_i F_\mu(\boldsymbol{\theta}_{t-1})\right|^2\right]\right) \\
&\leq \beta_2^t \boldsymbol{v}_{0,i} + (1-\beta_1)\Sigma^2 + \frac{\beta_1(1+\beta_1)^2 L^2 \eta^2 d}{(1-\beta_1)^2(1-\beta_2)} + \frac{(1+\beta_1)^2}{\beta_1}\sum_{\tau=1}^{t}(1-\beta_2)\beta_2^{t-\tau}\mathbb{E}\left[\left|\nabla_i F_\mu(\boldsymbol{\theta}_{\tau-1})\right|^2\right] .
\end{aligned}
\tag{36}
$$

### A.5. Proof of Cor. 5.5

Similar to the proof in Appx. A.4, let $\boldsymbol{g}_t = \hat{\nabla} f(\boldsymbol{\theta}_{t-1})$ we have

$$
\begin{aligned}
\mathbb{E}\left[\left|\boldsymbol{g}_{t,i}\right|^2\right] &= \mathbb{E}\left[\left|\boldsymbol{g}_{t,i} - \nabla_i F_\mu(\boldsymbol{\theta}_{t-1}) + \nabla_i F_\mu(\boldsymbol{\theta}_{t-1})\right|^2\right] \\
&\leq \left(1 + \frac{\beta_1}{1+\beta_1+\beta_1^2}\right)\mathbb{E}\left[\left|\boldsymbol{g}_{t,i} - \nabla_i F_\mu(\boldsymbol{\theta}_{t-1})\right|^2\right] + \left(1 + \frac{1+\beta_1+\beta_1^2}{\beta_1}\right)\mathbb{E}\left[\left|\nabla_i F_\mu(\boldsymbol{\theta}_{t-1})\right|^2\right] \\
&\leq \frac{(1+\beta_1)^2}{1+\beta_1+\beta_1^2}\Sigma^2 + \frac{(1+\beta_1)^2}{\beta_1}\mathbb{E}\left[\left|\nabla_i F_\mu(\boldsymbol{\theta}_{t-1})\right|^2\right] .
\end{aligned}
\tag{37}
$$

Consequently,

$$
\begin{aligned}
\mathbb{E}\left[\boldsymbol{v}_{t,i}\right] &= \mathbb{E}\left[\beta_2 \boldsymbol{v}_{t-1,i} + (1-\beta_2)\boldsymbol{g}_{t,i}^2\right] \\
&\leq \mathbb{E}[\beta_2 \boldsymbol{v}_{t-1,i}] + (1-\beta_2)\left(\frac{(1+\beta_1)^2}{1+\beta_1+\beta_1^2}\Sigma^2 + \frac{(1+\beta_1)^2}{\beta_1}\mathbb{E}\left[\left|\nabla_i F_\mu(\boldsymbol{\theta}_{t-1})\right|^2\right]\right) \\
&\leq \beta_2^t \boldsymbol{v}_{0,i} + \frac{(1+\beta_1)^2}{1+\beta_1+\beta_1^2}\Sigma^2 + \frac{(1+\beta_1)^2}{\beta_1}\sum_{\tau=1}^{t}(1-\beta_2)\beta_2^{t-\tau}\mathbb{E}\left[\left|\nabla_i F_\mu(\boldsymbol{\theta}_{\tau-1})\right|^2\right] \\
&\leq \beta_2^t \boldsymbol{v}_{0,i} + (1+\beta_1)\Sigma^2 + \frac{(1+\beta_1)^2}{\beta_1}\sum_{\tau=1}^{t}(1-\beta_2)\beta_2^{t-\tau}\mathbb{E}\left[\left|\nabla_i F_\mu(\boldsymbol{\theta}_{\tau-1})\right|^2\right] ,
\end{aligned}
\tag{38}
$$

which concludes our proof for Cor. 5.5.

## A.6. Proof of Lemma 5.6

By applying Hölder's inequality twice, we have the following

$$
\begin{aligned}
\left(\frac{1}{T}\sum_{t=0}^{T-1}\mathbb{E}\left[\|\nabla F_\mu(\boldsymbol{\theta}_t)\|\right]\right)^2 &= \frac{1}{T^2}\left(\sum_{t=0}^{T-1}\mathbb{E}\left[\sqrt[4]{\beta_2\|\boldsymbol{v}_t\|+\zeta}\frac{\|\nabla F_\mu(\boldsymbol{\theta}_t)\|}{\sqrt[4]{\beta_2\|\boldsymbol{v}_t\|+\zeta}}\right]\right)^2 \\
&\leq \frac{1}{T^2}\left(\sum_{t=0}^{T-1}\left(\mathbb{E}\left[\sqrt{\beta_2\|\boldsymbol{v}_t\|+\zeta}\right]\right)^{\frac{1}{2}}\left(\mathbb{E}\left[\frac{\|\nabla F_\mu(\boldsymbol{\theta}_t)\|^2}{\sqrt{\beta_2\|\boldsymbol{v}_t\|+\zeta}}\right]\right)^{\frac{1}{2}}\right)^2 \\
&\leq \left(\frac{1}{T}\sum_{t=0}^{T-1}\mathbb{E}\left[\sqrt{\beta_2\|\boldsymbol{v}_t\|+\zeta}\right]\right)\left(\frac{1}{T}\sum_{t=0}^{T-1}\mathbb{E}\left[\frac{\|\nabla F_\mu(\boldsymbol{\theta}_t)\|^2}{\sqrt{\beta_2\|\boldsymbol{v}_t\|+\zeta}}\right]\right),
\end{aligned}
\tag{39}
$$

which concludes our proof.

## A.7. Proof of Lemma. 5.7

Based on (34), (35), and (36), we have

$$
\begin{aligned}
\boldsymbol{v}_{t,i} &= \beta_2\boldsymbol{v}_{t-1,i} + (1-\beta_2)\boldsymbol{m}_{t,i}^2 \\
&\leq \beta_2\boldsymbol{v}_{t-1,i} + (1-\beta_2)\left((1+\beta_1)\left|\boldsymbol{m}_{t,i}-\nabla_iF_\mu(\boldsymbol{\theta}_{t-1})\right|^2 + \left(1+\frac{1}{\beta_1}\right)\left|\nabla_iF_\mu(\boldsymbol{\theta}_{t-1})\right|^2\right) \\
&= \beta_2\boldsymbol{v}_{t-1,i} + (1-\beta_2)\Big((1+\beta_1)\Big(\left|\boldsymbol{m}_{t,i}-\mathbb{E}\left[\boldsymbol{m}_{t,i}\right]\right|^2 + 2\left(\boldsymbol{m}_{t,i}-\mathbb{E}\left[\boldsymbol{m}_{t,i}\right]\right)\left(\mathbb{E}\left[\boldsymbol{m}_{t,i}\right]-\nabla_iF_\mu(\boldsymbol{\theta}_{t-1})\right) \\
&\quad + \left(1-\beta_1^t\right)\left|\frac{1}{1-\beta_1^t}\mathbb{E}[\boldsymbol{m}_{t,i}]-\nabla_iF_\mu(\boldsymbol{\theta}_{t-1})\right|^2 + \beta_1^t\left|\nabla_iF_\mu(\boldsymbol{\theta}_{t-1})\right|^2\Big) + \left(1+\frac{1}{\beta_1}\right)\left|\nabla_iF_\mu(\boldsymbol{\theta}_{t-1})\right|^2\Big) \\
&\leq \beta_2^t\boldsymbol{v}_{0,i} + \sum_{\tau=1}^t(1-\beta_2)\beta_2^{t-\tau}\Big((1+\beta_1)\Big(\left|\boldsymbol{m}_{t,i}-\mathbb{E}\left[\boldsymbol{m}_{t,i}\right]\right|^2 + 2\left(\boldsymbol{m}_{t,i}-\mathbb{E}\left[\boldsymbol{m}_{t,i}\right]\right)\left(\mathbb{E}\left[\boldsymbol{m}_{t,i}\right]-\nabla_iF_\mu(\boldsymbol{\theta}_{t-1})\right) \\
&\quad + \left(1-\beta_1^t\right)\left|\frac{1}{1-\beta_1^t}\mathbb{E}[\boldsymbol{m}_{t,i}]-\nabla_iF_\mu(\boldsymbol{\theta}_{t-1})\right|^2\Big) + \frac{(1+\beta_1)^2}{\beta_1}\left|\nabla_iF_\mu(\boldsymbol{\theta}_{t-1})\right|^2\Big)
\end{aligned}
\tag{40}
$$

Consequently,

$$
\begin{aligned}
&\frac{1}{T}\sum_{t=0}^{T-1}\mathbb{E}\left[\sqrt{\beta_2\|\boldsymbol{v}_t\|+\zeta}\right] \\
&\overset{(a)}{\leq} \sqrt{\zeta} + \frac{1}{T}\sum_{t=0}^{T-1}\sum_{i=1}^d\mathbb{E}\left[\sqrt{\beta_2\boldsymbol{v}_{t,i}}\right] \\
&\overset{(b)}{\leq} \sqrt{\zeta} + \frac{1}{T}\sum_{t=0}^{T-1}\sum_{i=1}^d\left(\sqrt{\beta_2}V + \frac{1+\beta_1}{\sqrt{\beta_1}}\sum_{\tau=1}^t\sqrt{1-\beta_2}\beta_2^{(t-\tau)/2}\mathbb{E}\left[\left|\nabla_iF_\mu(\boldsymbol{\theta}_{\tau-1})\right|\right]\right) \\
&\overset{(c)}{\leq} \sqrt{\zeta} + Vd + \frac{(1+\beta_1)\sqrt{d}}{\sqrt{\beta_1(1-\beta_2)}}\frac{1}{T}\sum_{t=0}^{T-1}\mathbb{E}\left[\|\nabla F_\mu(\boldsymbol{\theta}_t)\|\right]
\end{aligned}
\tag{41}
$$

where $(a)$, $(b)$ comes from the fact that $\sqrt{\sum_{i=1}^d a_i} \leq \sum_{i=1}^d\sqrt{a_i}$. In addition, $(b)$ also results from Jensen's inequality by getting the square root of $\left|\nabla_iF_\mu(\boldsymbol{\theta}_{t-1})\right|^2$ within the upper bound of $\boldsymbol{v}_{t,i}$ first (i.e., (40)) and taking expectation over it later, and $(c)$ is due to the sum of geometric series. Finally, $(c)$ is also the consequence of the sum of geometric series.

### A.8. Proof of Thm. 5.8

Inspired by the proof of Adam (Kingma & Ba, 2015) in FO optimization (Wang et al., 2024a; Zhang et al., 2024a), we focus on the study of the potential function $F_\mu(\boldsymbol{x}_t)$ with $\boldsymbol{x}_t$ defined as below:

$$\boldsymbol{x}_t \triangleq \frac{\boldsymbol{\theta}_t - \beta_1/\sqrt{\beta_2}\boldsymbol{\theta}_{t-1}}{1 - \beta_1/\sqrt{\beta_2}} \ . \tag{42}$$

Consequently,

$$\boldsymbol{x}_t - \boldsymbol{\theta}_t = \frac{\beta_1/\sqrt{\beta_2}}{1 - \beta_1/\sqrt{\beta_2}} \left(\boldsymbol{\theta}_t - \boldsymbol{\theta}_{t-1}\right) \ , \tag{43}$$

and

$$\boldsymbol{x}_{t+1} - \boldsymbol{x}_t = \frac{\boldsymbol{\theta}_{t+1} - \boldsymbol{\theta}_t - \beta_1/\sqrt{\beta_2}(\boldsymbol{\theta}_t - \boldsymbol{\theta}_{t-1})}{1 - \beta_1/\sqrt{\beta_2}}$$

$$= \frac{-\eta\boldsymbol{m}_{t+1}/\sqrt{\boldsymbol{v}_{t+1} + \zeta} + \eta\beta_1\boldsymbol{m}_t/\sqrt{\beta_2\boldsymbol{v}_t + \beta_2\zeta}}{1 - \beta_1/\sqrt{\beta_2}} \ . \tag{44}$$

According to the Lipschitz smoothness of function $F_\mu$, the following holds conditions on $\mathcal{F}_t$, i.e., the stochastics up to iteration $t$:

$$\mathbb{E}\left[F_\mu(\boldsymbol{x}_{t+1})|\mathcal{F}_t\right] \leq F_\mu(\boldsymbol{x}_t) + \mathbb{E}\left[\langle\nabla F_\mu(\boldsymbol{x}_t), \boldsymbol{x}_{t+1} - \boldsymbol{x}_t\rangle |\mathcal{F}_t\right] + \frac{\sqrt{d}L}{2}\mathbb{E}\left[\|\boldsymbol{x}_{t+1} - \boldsymbol{x}_t\|^2 |\mathcal{F}_t\right] \ . \tag{45}$$

We first reframe the second term on the RHS of (45) as below using the update rule in (7):

$$\mathbb{E}\left[\langle\nabla F_\mu(\boldsymbol{x}_t), \boldsymbol{x}_{t+1} - \boldsymbol{x}_t\rangle |\mathcal{F}_t\right]$$

$$= \mathbb{E}\left[\langle\nabla F_\mu(\boldsymbol{\theta}_t), \boldsymbol{x}_{t+1} - \boldsymbol{x}_t\rangle |\mathcal{F}_t\right] + \mathbb{E}\left[\langle\nabla F_\mu(\boldsymbol{x}_t) - \nabla F_\mu(\boldsymbol{\theta}_t), \boldsymbol{x}_{t+1} - \boldsymbol{x}_t\rangle |\mathcal{F}_t\right]$$

$$= \underbrace{\mathbb{E}\left[\left\langle\nabla F_\mu(\boldsymbol{\theta}_t), \frac{-\eta(1 - \beta_1)\hat{\nabla}f(\boldsymbol{\theta}_t, \xi_{t+1})}{(1 - \beta_1/\sqrt{\beta_2})\sqrt{\beta_2\boldsymbol{v}_t + \zeta}}\right\rangle \Big|\mathcal{F}_t\right]}_{①} + \underbrace{\mathbb{E}\left[\left\langle\nabla F_\mu(\boldsymbol{\theta}_t), \frac{-\eta\boldsymbol{m}_{t+1}/\sqrt{\boldsymbol{v}_{t+1} + \zeta} + \eta\boldsymbol{m}_{t+1}/\sqrt{\beta_2\boldsymbol{v}_t + \zeta}}{1 - \beta_1/\sqrt{\beta_2}}\right\rangle \Big|\mathcal{F}_t\right]}_{②}$$

$$+ \underbrace{\mathbb{E}\left[\left\langle\nabla F_\mu(\boldsymbol{\theta}_t), \frac{\eta\beta_1\boldsymbol{m}_t/\sqrt{\beta_2\boldsymbol{v}_t + \beta_2\zeta} - \eta\beta_1\boldsymbol{m}_t/\sqrt{\beta_2\boldsymbol{v}_t + \zeta}}{1 - \beta_1/\sqrt{\beta_2}}\right\rangle \Big|\mathcal{F}_t\right]}_{③} + \underbrace{\mathbb{E}\left[\langle\nabla F_\mu(\boldsymbol{x}_t) - \nabla F_\mu(\boldsymbol{\theta}_t), \boldsymbol{x}_{t+1} - \boldsymbol{x}_t\rangle |\mathcal{F}_t\right]}_{④} \tag{46}$$

We then bound the ①, ②, ③, and ④ term above separately. To begin with, we have the following based on the expectation of $\hat{\nabla}f(\boldsymbol{\theta}_t, \xi_{t+1})$:

$$① \triangleq \mathbb{E}\left[\mathbb{E}\left[\left\langle\nabla F_\mu(\boldsymbol{\theta}_t), \frac{-\eta(1 - \beta_1)\hat{\nabla}f(\boldsymbol{\theta}_t, \xi_{t+1})/\sqrt{\beta_2\boldsymbol{v}_t + \zeta}}{1 - \beta_1/\sqrt{\beta_2}}\right\rangle \Big|\mathcal{F}_t\right]\right]$$

$$= \frac{-\eta(1 - \beta_1)}{1 - \beta_1/\sqrt{\beta_2}}\sum_{i=1}^{d}\mathbb{E}\left[\frac{|\nabla_i F_\mu(\boldsymbol{\theta}_t)|^2}{\sqrt{\beta_2\boldsymbol{v}_{t,i} + \zeta}}\right] \ . \tag{47}$$

In addition, ② can be upper bounded as below:

$$
\begin{aligned}
② &\triangleq \mathbb{E}\left[\left\langle \nabla F_\mu(\boldsymbol{\theta}_t), \frac{-\eta \boldsymbol{m}_{t+1}/\sqrt{\boldsymbol{v}_{t+1}+\zeta} + \eta \boldsymbol{m}_{t+1}/\sqrt{\beta_2 \boldsymbol{v}_t + \zeta}}{1 - \beta_1/\sqrt{\beta_2}}\right\rangle \middle| \mathcal{F}_t\right] \\
&\overset{(a)}{\leq} \sum_{i=1}^d \frac{\eta}{1-\beta_1/\sqrt{\beta_2}} \mathbb{E}\left[\left|\nabla_i F_\mu(\boldsymbol{\theta}_t)\right| \frac{(1-\beta_2)\boldsymbol{m}_{t+1,i}^2\left|\boldsymbol{m}_{t+1,i}\right|}{\sqrt{\boldsymbol{v}_{t+1,i}+\zeta}\sqrt{\beta_2 \boldsymbol{v}_{t,i}+\zeta}(\sqrt{\beta_2 \boldsymbol{v}_{t,i}+\zeta}+\sqrt{\boldsymbol{v}_{t+1,i}+\zeta})}\middle|\mathcal{F}_t\right] \\
&\overset{(b)}{\leq} \sum_{i=1}^d \frac{\eta}{1-\beta_1/\sqrt{\beta_2}} \frac{\left|\nabla_i F_\mu(\boldsymbol{\theta}_t)\right|}{\sqrt{\beta_2 \boldsymbol{v}_{t,i}+\zeta}} \mathbb{E}\left[\frac{\sqrt{1-\beta_2}\boldsymbol{m}_{t+1,i}^2}{\sqrt{\beta_2 \boldsymbol{v}_{t,i}+\zeta}+\sqrt{\boldsymbol{v}_{t+1,i}+\zeta}}\middle|\mathcal{F}_t\right] \\
&\overset{(c)}{\leq} \sum_{i=1}^d \frac{\eta}{1-\beta_1/\sqrt{\beta_2}} \left(\frac{\left|\nabla_i F_\mu(\boldsymbol{\theta}_t)\right|^2}{2\gamma_0\sqrt{\beta_2 \boldsymbol{v}_{t,i}+\zeta}} + \frac{\gamma_0}{2\sqrt{\beta_2 \boldsymbol{v}_{t,i}+\zeta}}\left(\mathbb{E}\left[\frac{\sqrt{1-\beta_2}\boldsymbol{m}_{t+1,i}^2}{\sqrt{\beta_2 \boldsymbol{v}_{t,i}+\zeta}+\sqrt{\boldsymbol{v}_{t+1,i}+\zeta}}\middle|\mathcal{F}_t\right]\right)^2\right) \\
&\overset{(d)}{\leq} \sum_{i=1}^d \frac{\eta}{1-\beta_1/\sqrt{\beta_2}} \left(\frac{\left|\nabla_i F_\mu(\boldsymbol{\theta}_t)\right|^2}{2\gamma_0\sqrt{\beta_2 \boldsymbol{v}_{t,i}+\zeta}} + \frac{\gamma_0\mathbb{E}\left[\boldsymbol{m}_{t+1,i}^2|\mathcal{F}_t\right]}{2\sqrt{\beta_2 \boldsymbol{v}_{t,i}+\zeta}}\mathbb{E}\left[\frac{(1-\beta_2)\boldsymbol{m}_{t+1,i}^2}{(\sqrt{\beta_2 \boldsymbol{v}_{t,i}+\zeta}+\sqrt{\boldsymbol{v}_{t+1,i}+\zeta})^2}\middle|\mathcal{F}_t\right]\right) \quad (48) \\
&\overset{(e)}{\leq} \sum_{i=1}^d \frac{\eta}{1-\beta_1/\sqrt{\beta_2}} \left(\frac{\left|\nabla_i F_\mu(\boldsymbol{\theta}_t)\right|^2}{2\gamma_0\sqrt{\beta_2 \boldsymbol{v}_{t,i}+\zeta}} + \frac{\gamma_0\mathbb{E}\left[\boldsymbol{m}_{t+1,i}^2|\mathcal{F}_t\right]}{2}\right. \\
&\qquad\qquad \left. \times \mathbb{E}\left[\frac{\boldsymbol{v}_{t+1,i}-\beta_2\boldsymbol{v}_{t,i}}{\sqrt{\boldsymbol{v}_{t+1,i}+\zeta}\sqrt{\beta_2 \boldsymbol{v}_{t,i}+\zeta}\left(\sqrt{\beta_2 \boldsymbol{v}_{t,i}+\zeta}+\sqrt{\boldsymbol{v}_{t+1,i}+\zeta}\right)}\middle|\mathcal{F}_t\right]\right) \\
&\overset{(f)}{=} \sum_{i=1}^d \frac{\eta}{1-\beta_1/\sqrt{\beta_2}} \left(\frac{\left|\nabla_i F_\mu(\boldsymbol{\theta}_t)\right|^2}{2\gamma_0\sqrt{\beta_2 \boldsymbol{v}_{t,i}+\zeta}} + \frac{\gamma_0\mathbb{E}\left[\boldsymbol{m}_{t+1,i}^2|\mathcal{F}_t\right]}{2}\mathbb{E}\left[\frac{1}{\sqrt{\beta_2 \boldsymbol{v}_{t,i}+\zeta}} - \frac{1}{\sqrt{\boldsymbol{v}_{t+1,i}+\zeta}}\middle|\mathcal{F}_t\right]\right) \\
&\overset{(g)}{=} \sum_{i=1}^d \frac{\eta(1-\beta_1)}{4(1-\beta_1/\sqrt{\beta_2})} \frac{\left|\nabla_i F_\mu(\boldsymbol{\theta}_t)\right|^2}{\sqrt{\beta_2 \boldsymbol{v}_{t,i}+\zeta}} + \sum_{i=1}^d \frac{4\eta C^2 d^2}{(1-\beta_1/\sqrt{\beta_2})(1-\beta_1)\mu^2}\mathbb{E}\left[\frac{1}{\sqrt{\beta_2 \boldsymbol{v}_{t,i}+\zeta}} - \frac{1}{\sqrt{\boldsymbol{v}_{t+1,i}+\zeta}}\middle|\mathcal{F}_t\right]
\end{aligned}
$$

where $(a)$ is due to the update rule of second moment estimate in (6), $(b)$ results from $\frac{\left|\boldsymbol{m}_{t+1,i}\right|}{\sqrt{\boldsymbol{v}_{t+1,i}+\zeta}} \leq \frac{\left|\boldsymbol{m}_{t+1,i}\right|}{\sqrt{1-\beta_2}\left|\boldsymbol{m}_{t+1,i}\right|}$, $(c)$ is from $ab \leq \frac{a^2}{2\gamma_0} + \frac{\gamma_0 b^2}{2}$, $(d)$ is from the update rule in (6), $(f)$ can be obtained by choosing $\gamma_0 = \frac{2}{1-\beta_1}$ and $\sqrt{\beta_2 \boldsymbol{v}_{t,i}+\zeta} > \sqrt{\zeta}$, and $(g)$ results from (35) and (50) below.

$$
\begin{aligned}
\left|\hat{\nabla}_i f(\boldsymbol{\theta},\xi)\right| &= \left|\frac{d}{K}\sum_{k=1}^K \frac{f(\boldsymbol{\theta}+\mu\boldsymbol{u}_k;\xi) - f(\boldsymbol{\theta};\xi)}{\mu}\boldsymbol{u}_{k,i}\right| \\
&\leq \frac{d}{K}\sum_{k=1}^K \left|\frac{f(\boldsymbol{\theta}+\mu\boldsymbol{u}_k;\xi) - f(\boldsymbol{\theta};\xi)}{\mu}\right|\left|\boldsymbol{u}_{k,i}\right| \quad (49) \\
&\leq \frac{2Cd}{\mu},
\end{aligned}
$$

$$
\begin{aligned}
\left|\boldsymbol{m}_{t+1,i}\right| &= \left|(1-\beta_1)\sum_{\tau=1}^t \beta_1^{t-\tau}\hat{\nabla}_i f(\boldsymbol{\theta}_{\tau-1},\xi_\tau)\right| \\
&\leq (1-\beta_1)\sum_{\tau=1}^t \beta_1^{t-\tau}\left|\hat{\nabla}_i f(\boldsymbol{\theta}_{\tau-1},\xi_\tau)\right| \quad (50) \\
&\leq \frac{2Cd}{\mu}.
\end{aligned}
$$

Let $2\beta_2 \geq 1$, ③ can be bounded as below:

$$
\begin{aligned}
③ &\triangleq \mathbb{E}\left[\left\langle \nabla F_\mu(\boldsymbol{\theta}_t), \frac{\eta\beta_1 \boldsymbol{m}_t/\sqrt{\beta_2 \boldsymbol{v}_t + \beta_2\zeta} - \eta\beta_1 \boldsymbol{m}_t/\sqrt{\beta_2 \boldsymbol{v}_t + \zeta}}{1 - \beta_1/\sqrt{\beta_2}}\right\rangle \Big| \mathcal{F}_t\right] \\
&\overset{(a)}{=} \frac{\eta\beta_1}{1 - \beta_1/\sqrt{\beta_2}} \sum_{i=1}^d |\nabla_i F_\mu(\boldsymbol{\theta}_t)| \frac{(1-\beta_2)\zeta |\boldsymbol{m}_{t,i}|}{\sqrt{\beta_2 \boldsymbol{v}_{t,i} + \beta_2\zeta}\sqrt{\beta_2 \boldsymbol{v}_{t,i} + \zeta}(\sqrt{\beta_2 \boldsymbol{v}_{t,i} + \beta_2\zeta} + \sqrt{\beta_2 \boldsymbol{v}_{t,i} + \zeta})} \\
&\overset{(b)}{\leq} \frac{\eta\beta_1}{1 - \beta_1/\sqrt{\beta_2}} \sum_{i=1}^d |\nabla_i F_\mu(\boldsymbol{\theta}_t)| \frac{\sqrt{1-\beta_2}\sqrt{\zeta}}{\sqrt{\beta_2 \boldsymbol{v}_{t,i} + \zeta}} \\
&\overset{(c)}{\leq} \frac{\eta\beta_1\sqrt{\zeta}}{1 - \beta_1/\sqrt{\beta_2}} \sum_{i=1}^d \left(\frac{|\nabla_i F_\mu(\boldsymbol{\theta}_t)|^2}{2\gamma_1 \sqrt{\beta_2 \boldsymbol{v}_{t,i} + \zeta}} + \frac{\gamma_1(1-\beta_2)}{2\sqrt{\beta_2 \boldsymbol{v}_{t,i} + \zeta}}\right) \\
&\overset{(d)}{\leq} \frac{\eta\beta_1\sqrt{\zeta}}{1 - \beta_1/\sqrt{\beta_2}} \sum_{i=1}^d \frac{|\nabla_i F_\mu(\boldsymbol{\theta}_t)|^2}{2\gamma_1 \sqrt{\beta_2 \boldsymbol{v}_{t,i} + \zeta}} + \frac{\eta\beta_1\gamma_1(1-\beta_2)d}{2(1 - \beta_1/\sqrt{\beta_2})} \\
&\overset{(e)}{=} \frac{\eta(1-\beta_1)}{4(1 - \beta_1/\sqrt{\beta_2})} \sum_{i=1}^d \frac{|\nabla_i F_\mu(\boldsymbol{\theta}_t)|^2}{\sqrt{\beta_2 \boldsymbol{v}_{t,i} + \zeta}} + \frac{\eta\beta_1^2(1-\beta_2)d\sqrt{\zeta}}{(1 - \beta_1/\sqrt{\beta_2})(1-\beta_1)}
\end{aligned}
\tag{51}
$$

where $(b)$ results from $\frac{|\boldsymbol{m}_{t,i}|}{\sqrt{\boldsymbol{v}_{t,i}+\zeta}} \leq \frac{|\boldsymbol{m}_{t,i}|}{\sqrt{1-\beta_2}|\boldsymbol{m}_{t,i}|}$ and $2\beta_2 \geq 1$, $(c)$ is from $ab \leq \frac{a^2}{2\gamma_1} + \frac{\gamma_1 b^2}{2}$, and $(e)$ is obtained by choosing $\gamma_1 = \frac{2\beta_1\sqrt{\zeta}}{1-\beta_1}$.

Finally, ④ is bounded as below:

$$
\begin{aligned}
④ &\triangleq \mathbb{E}\left[\left\langle \nabla F_\mu(\boldsymbol{x}_t) - \nabla F_\mu(\boldsymbol{\theta}_t), \boldsymbol{x}_{t+1} - \boldsymbol{x}_t \right\rangle \big| \mathcal{F}_t\right] \\
&\overset{(a)}{\leq} \sum_{i=1}^d \frac{\beta_1 L/\sqrt{\beta_2}}{1 - \beta_1/\sqrt{\beta_2}} \mathbb{E}\left[\|\boldsymbol{\theta}_t - \boldsymbol{\theta}_{t-1}\| \left|\frac{\boldsymbol{\theta}_{t+1,i} - \boldsymbol{\theta}_{t,i} - \beta_1/\sqrt{\beta_2}(\boldsymbol{\theta}_{t,i} - \boldsymbol{\theta}_{t-1,i})}{1 - \beta_1/\sqrt{\beta_2}}\right| \Big| \mathcal{F}_t\right] \\
&\overset{(b)}{\leq} \sum_{i=1}^d \frac{\beta_1 L/\sqrt{\beta_2}}{(1 - \beta_1/\sqrt{\beta_2})^2} \mathbb{E}\left[\frac{\|\boldsymbol{\theta}_t - \boldsymbol{\theta}_{t-1}\|^2}{2\sqrt{d}} + \frac{\sqrt{d}|\boldsymbol{\theta}_{t+1,i} - \boldsymbol{\theta}_{t,i}|^2}{2} + \beta_1/\sqrt{\beta_2}\left(\frac{\|\boldsymbol{\theta}_t - \boldsymbol{\theta}_{t-1}\|^2}{2\sqrt{d}} + \frac{\sqrt{d}|\boldsymbol{\theta}_{t,i} - \boldsymbol{\theta}_{t-1,i}|^2}{2}\right) \Big| \mathcal{F}_t\right] \\
&\overset{(c)}{=} \frac{\beta_1 L\sqrt{d}/\sqrt{\beta_2}}{2(1 - \beta_1/\sqrt{\beta_2})^2} \left(\left(1 + 2\beta_1/\sqrt{\beta_2}\right)\mathbb{E}\left[\|\boldsymbol{\theta}_t - \boldsymbol{\theta}_{t-1}\|^2 \big| \mathcal{F}_t\right] + \mathbb{E}\left[\|\boldsymbol{\theta}_{t+1} - \boldsymbol{\theta}_t\|^2 \big| \mathcal{F}_t\right]\right) \\
&\overset{(d)}{=} \frac{\beta_1 L\eta^2\sqrt{d}/\sqrt{\beta_2}}{2(1 - \beta_1/\sqrt{\beta_2})^2} \sum_{i=1}^d \left(\left(1 + 2\beta_1/\sqrt{\beta_2}\right)\mathbb{E}\left[\frac{\boldsymbol{m}_{t,i}^2}{\boldsymbol{v}_{t,i} + \zeta}\Big| \mathcal{F}_t\right] + \mathbb{E}\left[\frac{\boldsymbol{m}_{t+1,i}^2}{\boldsymbol{v}_{t+1,i} + \zeta}\Big| \mathcal{F}_t\right]\right)
\end{aligned}
\tag{52}
$$

where $(a)$ is from (43), (44), Cauchy-Schwarz inequality, and the Lipschitz smoothness of $F_\mu$ in Lemma 5.2. In addition, $(b)$ is from $ab \leq \frac{a^2}{2\sqrt{d}} + \frac{\sqrt{d}b^2}{2}$, and $(d)$ is based on the update rule in (7).

we finally bound the last term on the RHS of (45) as below:

$$
\begin{aligned}
\frac{\sqrt{d}L}{2}\mathbb{E}\left[\|\boldsymbol{x}_{t+1} - \boldsymbol{x}_t\|^2 \big| \mathcal{F}_t\right] &= \sum_{i=1}^d \frac{\sqrt{d}L}{2(1 - \beta_1/\sqrt{\beta_2})^2} \mathbb{E}\left[\left|\boldsymbol{\theta}_{t+1,i} - \boldsymbol{\theta}_{t,i} - \beta_1/\sqrt{\beta_2}(\boldsymbol{\theta}_{t,i} - \boldsymbol{\theta}_{t-1,i})\right|^2 \big| \mathcal{F}_t\right] \\
&\leq \sum_{i=1}^d \frac{\sqrt{d}L}{2(1 - \beta_1/\sqrt{\beta_2})^2} \left(\mathbb{E}\left[2|\boldsymbol{\theta}_{t+1,i} - \boldsymbol{\theta}_{t,i}|^2 \big| \mathcal{F}_t\right] + 2\beta_1^2/\beta_2 \mathbb{E}\left[|\boldsymbol{\theta}_{t,i} - \boldsymbol{\theta}_{t-1,i}|^2 \big| \mathcal{F}_t\right]\right) \\
&= \sum_{i=1}^d \frac{\sqrt{d}L\eta^2}{(1 - \beta_1/\sqrt{\beta_2})^2} \left(\mathbb{E}\left[\frac{\boldsymbol{m}_{t+1,i}^2}{\boldsymbol{v}_{t+1,i} + \zeta}\Big| \mathcal{F}_t\right] + \beta_1^2/\beta_2 \mathbb{E}\left[\frac{\boldsymbol{m}_{t,i}^2}{\boldsymbol{v}_{t,i} + \zeta}\Big| \mathcal{F}_t\right]\right) .
\end{aligned}
\tag{53}
$$

By introducing (47), (48), (51), (52), (53) into (45), let $\beta_1 \leq \sqrt{\beta_2}$, $\boldsymbol{m}_{0,i} = 0$, $\boldsymbol{v}_{0,i} > 0$, we have the following

$$
\sum_{t=0}^{T-1} \left( \mathbb{E}\left[ F_\mu(\boldsymbol{x}_{t+1}) \right] - \mathbb{E}\left[ F_\mu(\boldsymbol{x}_t) \right] \right)
$$

$$
\leq - \frac{\eta(1 - \beta_1)}{2(1 - \beta_1/\sqrt{\beta_2})} \sum_{t=0}^{T-1} \sum_{i=1}^{d} \mathbb{E}\left[ \frac{|\nabla_i F_\mu(\boldsymbol{\theta}_t)|^2}{\sqrt{\beta_2 \boldsymbol{v}_{t,i}} + \zeta} \right] + \frac{3\sqrt{d}L\eta^2}{2(1 - \beta_1/\sqrt{\beta_2})^2} \sum_{t=0}^{T-1} \sum_{i=1}^{d} \mathbb{E}\left[ \frac{\boldsymbol{m}_{t+1,i}^2}{\boldsymbol{v}_{t+1,i} + \zeta} \right] +
$$

$$
\frac{5\sqrt{d}L\eta^2}{2(1 - \beta_1/\sqrt{\beta_2})^2} \sum_{t=0}^{T-1} \sum_{i=1}^{d} \mathbb{E}\left[ \frac{\boldsymbol{m}_{t,i}^2}{\boldsymbol{v}_{t,i} + \zeta} \right] + \sum_{t=0}^{T-1} \sum_{i=1}^{d} \frac{4\eta C^2 d^2}{(1 - \beta_1/\sqrt{\beta_2})(1 - \beta_1)\mu^2} \mathbb{E}\left[ \frac{1}{\sqrt{\beta_2 \boldsymbol{v}_{t,i}} + \zeta} - \frac{1}{\sqrt{\boldsymbol{v}_{t+1,i}} + \zeta} \right] +
$$

$$
T \frac{\eta\beta_1^2(1 - \beta_2)d\sqrt{\zeta}}{(1 - \beta_1/\sqrt{\beta_2})(1 - \beta_1)}
$$

$$
\leq - \frac{\eta(1 - \beta_1)}{2(1 - \beta_1/\sqrt{\beta_2})} \sum_{t=0}^{T-1} \sum_{i=1}^{d} \mathbb{E}\left[ \frac{|\nabla_i F_\mu(\boldsymbol{\theta}_t)|^2}{\sqrt{\beta_2 \boldsymbol{v}_t} + \zeta} \right] + \frac{4\sqrt{d}L\eta^2}{(1 - \beta_1/\sqrt{\beta_2})^2} \sum_{i=1}^{d} \left( \frac{\ln\left( \left( \beta_2^T \boldsymbol{v}_{0,i} + 4C^2 d^2/\mu^2 \right)/\boldsymbol{v}_{0,i} \right)}{1 - \beta_2} + 2T \right) +
$$

$$
\frac{4\eta C^2 d^3}{(1 - \beta_1/\sqrt{\beta_2})(1 - \beta_1)\mu^2} \left( \frac{1}{\sqrt{\zeta}} + \frac{T(1 - \beta_2)}{\sqrt{\zeta}} \right) + T \frac{\eta\beta_1^2(1 - \beta_2)d\sqrt{\zeta}}{(1 - \beta_1/\sqrt{\beta_2})(1 - \beta_1)}
$$

$$
\tag{54}
$$

where the last inequality comes from the following (55) and (57).

$$
\sum_{t=0}^{T-1} \mathbb{E}\left[ \frac{1}{\sqrt{\beta_2 \boldsymbol{v}_{t,i}} + \zeta} - \frac{1}{\sqrt{\boldsymbol{v}_{t+1,i}} + \zeta} \right] = \frac{1}{\sqrt{\beta_2 \boldsymbol{v}_{0,i}} + \zeta} + \sum_{t=0}^{T-2} \mathbb{E}\left[ \frac{1}{\sqrt{\beta_2 \boldsymbol{v}_{t+1,i}} + \zeta} - \frac{1}{\sqrt{\boldsymbol{v}_{t+1,i}} + \zeta} \right] - \mathbb{E}\left[ \frac{1}{\sqrt{\boldsymbol{v}_{T,i}} + \zeta} \right]
$$

$$
\leq \frac{1}{\sqrt{\zeta}} + \sum_{t=1}^{T-1} \mathbb{E}\left[ \frac{1}{\sqrt{\beta_2 \boldsymbol{v}_{t+1,i}} + \zeta} - \frac{\sqrt{\beta_2}}{\sqrt{\beta_2 \boldsymbol{v}_{t+1,i}} + \zeta} \right]
$$

$$
\leq \frac{1}{\sqrt{\zeta}} + \frac{T(1 - \beta_2)}{\sqrt{\zeta}} .
$$

$$
\tag{55}
$$

Moreover, due to the fact that $\frac{a}{1+a} \leq \ln(1 + a)$, the following holds:

$$
\frac{(1 - \beta_2)\boldsymbol{m}_{t,i}^2}{\boldsymbol{v}_{t,i}} = \frac{\frac{(1-\beta_2)\boldsymbol{m}_{t,i}^2}{\boldsymbol{v}_{t,i} - (1-\beta_2)\boldsymbol{m}_{t,i}^2}}{1 + \frac{(1-\beta_2)\boldsymbol{m}_{t,i}^2}{\boldsymbol{v}_{t,i} - (1-\beta_2)\boldsymbol{m}_{t,i}^2}} \leq \ln\left( 1 + \frac{(1 - \beta_2)\boldsymbol{m}_{t,i}^2}{\boldsymbol{v}_{t,i} - (1 - \beta_2)\boldsymbol{m}_{t,i}^2} \right) = \ln\left( \frac{\boldsymbol{v}_{t,i}}{\beta_2 \boldsymbol{v}_{t-1,i}} \right) = \ln\left( \frac{\boldsymbol{v}_{t,i}}{\boldsymbol{v}_{t-1,i}} \right) - \ln(\beta_2) .
$$

$$
\tag{56}
$$

Given the results above and $2\beta_2 \geq 1$, we have

$$
\mathbb{E}\left[ \sum_{t=0}^{T} \frac{\boldsymbol{m}_{t,i}^2}{\boldsymbol{v}_{t,i} + \zeta} \right] \leq \frac{1}{1 - \beta_2} \left( \mathbb{E}\left[ \ln(\boldsymbol{v}_{T,i}) - \ln(\boldsymbol{v}_{0,i}) \right] - T\ln\beta_2 \right)
$$

$$
\leq \frac{1}{1 - \beta_2} \left( \ln\left( \mathbb{E}\left[ \frac{\boldsymbol{v}_{T,i}}{\boldsymbol{v}_{0,i}} \right] \right) + 2T(1 - \beta_2) \right)
$$

$$
\leq \frac{1}{1 - \beta_2} \ln\left( \frac{\beta_2^T \boldsymbol{v}_{0,i} + 4C^2 d^2/\mu^2}{\boldsymbol{v}_{0,i}} \right) + 2T
$$

$$
\tag{57}
$$

where the second inequality is due to $\ln a \leq a - 1$ and last inequality comes from (50).

Define $\Delta \triangleq F_\mu(\boldsymbol{x}_1) - F_\mu^*$, by re-arranging (54), we have

$$
\frac{1}{T}\sum_{t=0}^{T-1}\sum_{i=1}^{d}\mathbb{E}\left[\frac{|\nabla_i F_\mu(\boldsymbol{\theta}_t)|^2}{\sqrt{\beta_2 \boldsymbol{v}_{t,i}+\zeta}}\right] \leq \frac{2(1-\beta_1/\sqrt{\beta_2})\Delta}{\eta T(1-\beta_1)} + \frac{8L\eta\sqrt{d}}{(1-\beta_1/\sqrt{\beta_2})(1-\beta_1)}\sum_{i=1}^{d}\left(\frac{\ln\left(\left(\beta_2^T \boldsymbol{v}_{0,i}+4C^2 d^2/\mu^2\right)/\boldsymbol{v}_{0,i}\right)}{T(1-\beta_2)}+2\right) +
$$
$$
\frac{8C^2 d^3}{(1-\beta_1)^2\mu^2}\left(\frac{1}{T\sqrt{\zeta}}+\frac{1-\beta_2}{\sqrt{\zeta}}\right) + \frac{2\beta_1^2(1-\beta_2)d\sqrt{\zeta}}{(1-\beta_1)^2}.
\tag{58}
$$

By choosing $1-\beta_2 = \left(\frac{8C^2 d^3}{(1-\beta_1)^2\mu^2\sqrt{\zeta}}+\frac{2\beta_1^2 d\sqrt{\zeta}}{(1-\beta_1)^2}\right)^{-1}\epsilon^2/3 \sim \mathcal{O}(\epsilon^2), \eta = \left(\frac{16Ld\sqrt{d}}{(1-\beta_1/\sqrt{\beta_2})(1-\beta_1)}\right)^{-1}\epsilon^2/3 \sim \mathcal{O}(\epsilon^2)$ and $T = 3\epsilon^{-2}\left(\frac{2(1-\beta_1/\sqrt{\beta_2})\Delta}{\eta(1-\beta_1)}+\frac{8C^2 d^3}{(1-\beta_1)^2\mu^2\sqrt{\xi}}+\frac{8L\eta\sqrt{d}}{(1-\beta_1/\sqrt{\beta_2})(1-\beta_1)}\sum_{i=1}^{d}\left(\frac{\ln\left((\beta_2^T \boldsymbol{v}_{0,i}+4C^2 d^2/\mu^2)/\boldsymbol{v}_{0,i}\right)}{(1-\beta_2)}\right)\right) \sim \mathcal{O}(\epsilon^{-4})$, we can simply have the following,

$$
\frac{1}{T}\sum_{t=0}^{T-1}\mathbb{E}\left[\frac{\|\nabla F_\mu(\boldsymbol{\theta}_t)\|^2}{\sqrt{\beta_2\|\boldsymbol{v}_t\|+\zeta}}\right] \leq \frac{1}{T}\sum_{t=0}^{T-1}\sum_{i=1}^{d}\mathbb{E}\left[\frac{|\nabla_i F_\mu(\boldsymbol{\theta}_t)|^2}{\sqrt{\beta_2 \boldsymbol{v}_{t,i}+\zeta}}\right] \leq \epsilon^2
\tag{59}
$$

where the first inequality is due to the fact that $\sum_i\left(a_i/\sum_j b_j\right) = \sum_i a_i/\sum_i b_i \leq \sum a_i/b_i$. We therefore conclude our proof of Thm. 5.8.

## A.9. Proof of Thm. 5.9

By introducing (41) and (59) into Lemma 5.6, we have

$$
\left(\frac{1}{T}\sum_{t=0}^{T-1}\mathbb{E}\left[\|\nabla F_\mu(\boldsymbol{\theta}_t)\|\right]\right)^2 \leq \left(\frac{1}{T}\sum_{t=0}^{T-1}\mathbb{E}\left[\sqrt{\beta_2\|\boldsymbol{v}_t\|+\zeta}\right]\right)\left(\frac{1}{T}\sum_{t=0}^{T-1}\mathbb{E}\left[\frac{\|\nabla F_\mu(\boldsymbol{\theta}_t)\|^2}{\sqrt{\beta_2\|\boldsymbol{v}_t\|+\zeta}}\right]\right)
$$
$$
\leq \left(\sqrt{\zeta}+Vd+\frac{(1+\beta_1)\sqrt{d}}{\sqrt{\beta_1(1-\beta_2)}}\frac{1}{T}\sum_{t=0}^{T-1}\mathbb{E}\left[\|\nabla F_\mu(\boldsymbol{\theta}_t)\|\right]\right)\epsilon^2.
\tag{60}
$$

By applying the formula for the root of square equation, we have the following

$$
\frac{1}{T}\sum_{t=0}^{T-1}\mathbb{E}\left[\|\nabla F_\mu(\boldsymbol{\theta}_t)\|\right] \leq \frac{(1+\beta_1)\sqrt{d}}{\sqrt{\beta_1(1-\beta_2)}}\epsilon^2 + \left(\sqrt[4]{\zeta}+\sqrt{Vd}\right)\epsilon,
\tag{61}
$$

which concludes our proof for Thm. 5.9.

## A.10. Proof of Thm. 5.10

We first introduce Lemma A.1 and Lemma A.2 from (Wang et al., 2024a):

**Lemma A.1** (Lemma 6 in (Wang et al., 2024a))**.** *The following inequality holds for ZO-AdaMM if* $0 < \beta_1^2 < \beta_2 < 1$,

$$
\frac{\boldsymbol{m}_{t,i}}{\sqrt{\boldsymbol{v}_{t,i}+\zeta}} \leq \frac{1-\beta_1}{\sqrt{1-\beta_2}\sqrt{1-\beta_1^2/\beta_2}}.
\tag{62}
$$

*Proof.* Let $\boldsymbol{g}_{\tau,i} \triangleq \hat{\nabla}_i f(\boldsymbol{\theta}_\tau, \xi_{\tau+1})$, we have

$$
\begin{aligned}
\frac{\boldsymbol{m}_{t,i}}{\sqrt{\boldsymbol{v}_{t,i} + \zeta}} &\leq \frac{\boldsymbol{m}_{t,i}}{\sqrt{\boldsymbol{v}_{t,i}}} = \frac{(1-\beta_1) \sum_{\tau=0}^{t-1} \beta_1^{t-\tau} \boldsymbol{g}_{\tau,i}}{\sqrt{(1-\beta_2) \sum_{\tau=0}^{t-1} \beta_2^{t-\tau} \boldsymbol{g}_{\tau,i}^2}} \\
&\overset{(a)}{\leq} \frac{1-\beta_1}{\sqrt{1-\beta_2}} \frac{\sqrt{\sum_{\tau=0}^{t-1} \beta_2^{t-\tau} \boldsymbol{g}_{\tau,i}^2} \sqrt{\sum_{\tau=0}^{t-1} \left(\beta_1^2/\beta_2\right)^{t-\tau}}}{\sqrt{\sum_{\tau=0}^{t-1} \beta_2^{t-\tau} \boldsymbol{g}_{\tau,i}^2}} \\
&\leq \frac{1-\beta_1}{\sqrt{1-\beta_2}\sqrt{1-\left(\beta_1^2/\beta_2\right)}}
\end{aligned}
\tag{63}
$$

where $(a)$ is due to the Cauchy-Schwarz inequality. $\qquad\square$

**Lemma A.2** (Lemma 5 in (Wang et al., 2024a)). *The following inequality holds for ZO-AdaMM if $0 < \beta_1^2 < \beta_2 < 1$,*

$$
\sum_{t=0}^{T-1} \frac{(1-\beta_2)\boldsymbol{m}_{t,i}^2}{\boldsymbol{v}_{t,i}+\zeta} \leq \frac{(1-\beta_1)^2}{\left(1-\beta_1/\sqrt{\beta_2}\right)^2} \left(\ln\left(\frac{\boldsymbol{v}_T}{\boldsymbol{v}_0}\right) - T\ln\beta_2\right).
\tag{64}
$$

*Proof.* Let $\boldsymbol{g}_{\tau,i} \triangleq \hat{\nabla}_i f(\boldsymbol{\theta}_\tau, \xi_{\tau+1})$, similar to (56), we have,

$$
\sum_{t=0}^{T-1} \frac{(1-\beta_2)\boldsymbol{g}_{t,i}^2}{\boldsymbol{v}_t + \zeta} \leq \ln\left(\frac{\boldsymbol{v}_T}{\boldsymbol{v}_0}\right) - T\ln\beta_2.
\tag{65}
$$

Besides,

$$
\frac{\boldsymbol{m}_{t,i}}{\sqrt{\boldsymbol{v}_{t,i}+\zeta}} \leq \frac{\boldsymbol{m}_{t,i}}{\sqrt{\boldsymbol{v}_{t,i}}} \leq (1-\beta_1) \sum_{\tau=0}^{t} \beta_1^{t-\tau} \frac{\boldsymbol{g}_{\tau,i}}{\sqrt{\boldsymbol{v}_{t,i}}} \leq (1-\beta_1) \sum_{\tau=0}^{t} \left(\frac{\beta_1}{\sqrt{\beta_2}}\right)^{t-\tau} \frac{\boldsymbol{g}_{\tau,i}}{\sqrt{\boldsymbol{v}_{\tau,i}}}.
\tag{66}
$$

Therefore, we have,

$$
\begin{aligned}
\sum_{t=0}^{T-1} \frac{(1-\beta_2)\boldsymbol{m}_{t,i}^2}{\boldsymbol{v}_{t,i}+\zeta} &\leq (1-\beta_2)(1-\beta_1)^2 \sum_{t=0}^{T-1} \left(\sum_{\tau=0}^{t} \left(\frac{\beta_1}{\sqrt{\beta_2}}\right)^{t-\tau} \frac{\boldsymbol{g}_{\tau,i}}{\sqrt{\boldsymbol{v}_{\tau,i}}}\right)^2 \\
&\overset{(a)}{\leq} (1-\beta_2)(1-\beta_1)^2 \sum_{t=0}^{T-1} \left(\sum_{\tau=0}^{t} \left(\frac{\beta_1}{\sqrt{\beta_2}}\right)^{t-\tau}\right) \left(\sum_{\tau=0}^{t} \left(\frac{\beta_1}{\sqrt{\beta_2}}\right)^{t-\tau} \frac{\boldsymbol{g}_{\tau,i}^2}{\boldsymbol{v}_{\tau,i}}\right) \\
&\leq (1-\beta_2)\frac{(1-\beta_1)^2}{1-\beta_1/\sqrt{\beta_2}} \sum_{t=0}^{T-1}\sum_{\tau=0}^{t} \left(\frac{\beta_1}{\sqrt{\beta_2}}\right)^{t-\tau} \frac{\boldsymbol{g}_{\tau,i}^2}{\boldsymbol{v}_{\tau,i}} \\
&\overset{(b)}{=} (1-\beta_2)\frac{(1-\beta_1)^2}{1-\beta_1/\sqrt{\beta_2}} \sum_{\tau=0}^{T-1}\sum_{t=\tau}^{T-1} \left(\frac{\beta_1}{\sqrt{\beta_2}}\right)^{t-\tau} \frac{\boldsymbol{g}_{\tau,i}^2}{\boldsymbol{v}_{\tau,i}} \\
&\leq \frac{(1-\beta_1)^2}{\left(1-\beta_1/\sqrt{\beta_2}\right)^2} \sum_{t=0}^{T-1} \frac{(1-\beta_2)\boldsymbol{g}_{t,i}^2}{\boldsymbol{v}_{t,i}} \\
&\overset{(c)}{\leq} \frac{(1-\beta_1)^2}{\left(1-\beta_1/\sqrt{\beta_2}\right)^2} \left(\ln\left(\frac{\boldsymbol{v}_T}{\boldsymbol{v}_0}\right) - T\ln\beta_2\right)
\end{aligned}
\tag{67}
$$

where $(a)$ comes from Cauchy-Schwarz inequality and $(c)$ is due to (65). In $(b)$ we exchange the order of summation over $t$ and $\tau$. $\qquad\square$

Following the proof idea in Appx. A.7, we have the following for ZO-AdaMM,

$$
\begin{aligned}
&\frac{1}{T}\sum_{t=0}^{T-1}\mathbb{E}\left[\sqrt{\beta_2\left\|\boldsymbol{v}_t\right\|}+\zeta\right]\\
&\stackrel{(a)}{\leq}\sqrt{\zeta}+\frac{1}{T}\sum_{t=0}^{T-1}\sum_{i=1}^{d}\mathbb{E}\left[\sqrt{\beta_2\boldsymbol{v}_{t,i}}\right]\\
&\stackrel{(b)}{\leq}\sqrt{\zeta}+\frac{1}{T}\sum_{t=0}^{T-1}\sum_{i=1}^{d}\left(\sqrt{\beta_2}\hat{V}+\frac{1+\beta_1}{\sqrt{\beta_1}}\sum_{\tau=1}^{t}\sqrt{1-\beta_2}\beta_2^{(t-\tau)/2}\mathbb{E}\left[\left|\nabla_i F_\mu(\boldsymbol{\theta}_{\tau-1})\right|\right]\right)\\
&\stackrel{(c)}{\leq}\sqrt{\zeta}+\hat{V}d+\frac{(1+\beta_1)\sqrt{d}}{\sqrt{\beta_1(1-\beta_2)}}\frac{1}{T}\sum_{t=0}^{T-1}\mathbb{E}\left[\left\|\nabla F_\mu(\boldsymbol{\theta}_t)\right\|\right]
\end{aligned}
\tag{68}
$$

where $\hat{V}^2\triangleq\left\|\boldsymbol{v}_0\right\|+(1+\beta_1)\frac{8(\sigma^2+C^2)d}{K\mu^2}$.

In addition, following the proof idea in Appx. A.8, we have

$$
\mathbb{E}\left[\left\langle\nabla F_\mu(\boldsymbol{x}_t),\boldsymbol{x}_{t+1}-\boldsymbol{x}_t\right\rangle|\mathcal{F}_t\right]
$$

$$
=\underbrace{\mathbb{E}\left[\left\langle\nabla F_\mu(\boldsymbol{\theta}_t),\frac{-\eta(1-\beta_1)\hat{\nabla}f(\boldsymbol{\theta}_t,\xi_{t+1})}{(1-\beta_1/\sqrt{\beta_2})\sqrt{\beta_2\boldsymbol{v}_t}+\zeta}\right\rangle\bigg|\mathcal{F}_t\right]}_{①}+\underbrace{\mathbb{E}\left[\left\langle\nabla F_\mu(\boldsymbol{\theta}_t),\frac{-\eta\boldsymbol{m}_{t+1}/\sqrt{\boldsymbol{v}_{t+1}+\zeta}+\eta\boldsymbol{m}_{t+1}/\sqrt{\beta_2\boldsymbol{v}_t}+\zeta}{1-\beta_1/\sqrt{\beta_2}}\right\rangle\bigg|\mathcal{F}_t\right]}_{②}
$$

$$
+\underbrace{\mathbb{E}\left[\left\langle\nabla F_\mu(\boldsymbol{\theta}_t),\frac{\eta\beta_1\boldsymbol{m}_t/\sqrt{\beta_2\boldsymbol{v}_t+\beta_2\zeta}-\eta\beta_1\boldsymbol{m}_t/\sqrt{\beta_2\boldsymbol{v}_t}+\zeta}{1-\beta_1/\sqrt{\beta_2}}\right\rangle\bigg|\mathcal{F}_t\right]}_{③}+\underbrace{\mathbb{E}\left[\left\langle\nabla F_\mu(\boldsymbol{x}_t)-\nabla F_\mu(\boldsymbol{\theta}_t),\boldsymbol{x}_{t+1}-\boldsymbol{x}_t\right\rangle|\mathcal{F}_t\right]}_{④}
\tag{69}
$$

where each item can be upper bounded as below.

$$
\begin{aligned}
①&=\frac{-\eta(1-\beta_1)}{1-\beta_1/\sqrt{\beta_2}}\sum_{i=1}^{d}\mathbb{E}\left[\frac{\left|\nabla_i F_\mu(\boldsymbol{\theta}_t)\right|^2}{\sqrt{\beta_2\boldsymbol{v}_{t,i}}+\zeta}\right],\\
②&\leq\sum_{i=1}^{d}\frac{\eta(1-\beta_1)}{4(1-\beta_1/\sqrt{\beta_2})}\frac{\left|\nabla_i F_\mu(\boldsymbol{\theta}_t)\right|^2}{\sqrt{\beta_2\boldsymbol{v}_{t,i}}+\zeta}+\sum_{i=1}^{d}\frac{4\eta(1-\beta_1)C^2d^2}{(1-\beta_1/\sqrt{\beta_2})(1-\beta_1^2/\beta_2)\mu^2}\mathbb{E}\left[\left|\frac{1}{\sqrt{\beta_2\boldsymbol{v}_{t,i}}+\zeta}-\frac{1}{\sqrt{\boldsymbol{v}_{t+1,i}}+\zeta}\right|\bigg|\mathcal{F}_t\right],\\
③&\leq\frac{\eta(1-\beta_1)}{4(1-\beta_1/\sqrt{\beta_2})}\sum_{i=1}^{d}\frac{\left|\nabla_i F_\mu(\boldsymbol{\theta}_t)\right|^2}{\sqrt{\beta_2\boldsymbol{v}_{t,i}}+\zeta}+\frac{\eta\beta_1^2(1-\beta_1)(1-\beta_2)d\sqrt{\zeta}}{(1-\beta_1/\sqrt{\beta_2})(1-\beta_1^2/\beta_2)},\\
④&\leq\frac{\beta_1 L\eta^2\sqrt{d}/\sqrt{\beta_2}}{2(1-\beta_1/\sqrt{\beta_2})^2}\sum_{i=1}^{d}\left(\left(1+2\beta_1/\sqrt{\beta_2}\right)\mathbb{E}\left[\frac{\boldsymbol{m}_{t,i}^2}{\boldsymbol{v}_{t,i}+\zeta}\bigg|\mathcal{F}_t\right]+\mathbb{E}\left[\frac{\boldsymbol{m}_{t+1,i}^2}{\boldsymbol{v}_{t+1,i}+\zeta}\bigg|\mathcal{F}_t\right]\right).
\end{aligned}
\tag{70}
$$

Here, both ② and ③ results from Lemma A.1.

Finally, we have

$$
\sum_{t=0}^{T-1} \left( \mathbb{E}\left[ F_\mu(\boldsymbol{x}_{t+1}) \right] - \mathbb{E}\left[ F_\mu(\boldsymbol{x}_t) \right] \right)
$$

$$
\leq -\frac{\eta(1-\beta_1)}{2(1-\beta_1/\sqrt{\beta_2})} \sum_{t=0}^{T-1} \sum_{i=1}^{d} \mathbb{E}\left[ \frac{\left|\nabla_i F_\mu(\boldsymbol{\theta}_t)\right|^2}{\sqrt{\beta_2 \boldsymbol{v}_t} + \zeta} \right] + \frac{4\sqrt{d}L\eta^2(1-\beta_1)^2}{(1-\beta_1/\sqrt{\beta_2})^4} \sum_{i=1}^{d} \left( \frac{\ln\left( \left( \beta_2^T \boldsymbol{v}_{0,i} + 4C^2 d^2/\mu^2 \right)/\boldsymbol{v}_{0,i} \right)}{1-\beta_2} + 2T \right) +
$$

$$
\frac{4\eta(1-\beta_1)C^2 d^3}{(1-\beta_1/\sqrt{\beta_2})(1-\beta_1^2/\beta_2)\mu^2} \left( \frac{1}{\sqrt{\zeta}} + \frac{T(1-\beta_2)}{\sqrt{\zeta}} \right) + T\frac{\eta\beta_1^2(1-\beta_1)(1-\beta_2)d\sqrt{\zeta}}{(1-\beta_1/\sqrt{\beta_2})(1-\beta_1^2/\beta_2)} \,.
\tag{71}
$$

Define $\Delta \triangleq F_\mu(\boldsymbol{x}_1) - F_\mu^*$, by re-arranging the result above, we have

$$
\frac{1}{T} \sum_{t=0}^{T-1} \sum_{i=1}^{d} \mathbb{E}\left[ \frac{\left|\nabla_i F_\mu(\boldsymbol{\theta}_t)\right|^2}{\sqrt{\beta_2 \boldsymbol{v}_{t,i}} + \zeta} \right] \leq \frac{2(1-\beta_1/\sqrt{\beta_2})\Delta}{\eta T(1-\beta_1)} + \frac{8L\eta\sqrt{d}(1-\beta_1)}{(1-\beta_1/\sqrt{\beta_2})^3} \sum_{i=1}^{d} \left( \frac{\ln\left( \left( \beta_2^T \boldsymbol{v}_{0,i} + 4C^2 d^2/\mu^2 \right)/\boldsymbol{v}_{0,i} \right)}{T(1-\beta_2)} + 2 \right) +
$$

$$
\frac{8C^2 d^3}{(1-\beta_1^2/\beta_2)\mu^2} \left( \frac{1}{T\sqrt{\zeta}} + \frac{1-\beta_2}{\sqrt{\zeta}} \right) + \frac{2\beta_1^2(1-\beta_2)d\sqrt{\zeta}}{1-\beta_1^2/\beta_2} \,.
\tag{72}
$$

By choosing $1 - \beta_2 = \left( \frac{8C^2 d^3}{(1-\beta_1^2/\beta_2)\mu^2\sqrt{\zeta}} + \frac{2\beta_1^2 d\sqrt{\zeta}}{1-\beta_1^2/\beta_2} \right)^{-1} \epsilon^2/3 \sim \mathcal{O}(\epsilon^2), \eta = \left( \frac{16Ld\sqrt{d}(1-\beta_1)}{(1-\beta_1/\sqrt{\beta_2})^3} \right)^{-1} \epsilon^2/3 \sim \mathcal{O}(\epsilon^2)$ and $T = 3\epsilon^{-2} \left( \frac{2(1-\beta_1/\sqrt{\beta_2})\Delta}{\eta(1-\beta_1)} + \frac{8C^2 d^3}{(1-\beta_1^2/\beta_2)\mu^2\sqrt{\xi}} + \frac{8L\eta\sqrt{d}(1-\beta_1)}{(1-\beta_1/\sqrt{\beta_2})^3} \sum_{i=1}^{d} \left( \frac{\ln\left( \left( \beta_2^T \boldsymbol{v}_{0,i} + 4C^2 d^2/\mu^2 \right)/\boldsymbol{v}_{0,i} \right)}{(1-\beta_2)} \right) \right) \sim \mathcal{O}(\epsilon^{-4})$, we can simply have the following,

$$
\frac{1}{T} \sum_{t=0}^{T-1} \mathbb{E}\left[ \frac{\left\|\nabla F_\mu(\boldsymbol{\theta}_t)\right\|^2}{\sqrt{\beta_2 \left\|\boldsymbol{v}_t\right\|} + \zeta} \right] \leq \frac{1}{T} \sum_{t=0}^{T-1} \sum_{i=1}^{d} \mathbb{E}\left[ \frac{\left|\nabla_i F_\mu(\boldsymbol{\theta}_t)\right|^2}{\sqrt{\beta_2 \boldsymbol{v}_{t,i}} + \zeta} \right] \leq \epsilon^2 \,.
\tag{73}
$$

By introducing (68) and (73) into Lemma 5.6, we have

$$
\left( \frac{1}{T} \sum_{t=0}^{T-1} \mathbb{E}\left[ \left\|\nabla F_\mu(\boldsymbol{\theta}_t)\right\| \right] \right)^2 \leq \left( \frac{1}{T} \sum_{t=0}^{T-1} \mathbb{E}\left[ \sqrt{\beta_2 \left\|\boldsymbol{v}_t\right\|} + \zeta \right] \right) \left( \frac{1}{T} \sum_{t=0}^{T-1} \mathbb{E}\left[ \frac{\left\|\nabla F_\mu(\boldsymbol{\theta}_t)\right\|^2}{\sqrt{\beta_2 \left\|\boldsymbol{v}_t\right\|} + \zeta} \right] \right)
$$

$$
\leq \left( \sqrt{\zeta} + \hat{V}d + \frac{(1+\beta_1)\sqrt{d}}{\sqrt{\beta_1(1-\beta_2)}} \frac{1}{T} \sum_{t=0}^{T-1} \mathbb{E}\left[ \left\|\nabla F_\mu(\boldsymbol{\theta}_t)\right\| \right] \right) \epsilon^2 \,.
\tag{74}
$$

By applying the formula for the root of square equation, we have the following

$$
\frac{1}{T} \sum_{t=0}^{T-1} \mathbb{E}\left[ \left\|\nabla F_\mu(\boldsymbol{\theta}_t)\right\| \right] \leq \frac{(1+\beta_1)\sqrt{d}}{\sqrt{\beta_1(1-\beta_2)}} \epsilon^2 + \left( \sqrt[4]{\zeta} + \sqrt{\hat{V}d} \right) \epsilon \,,
\tag{75}
$$

which concludes our proof for Thm. 5.10.

# B. Experiments

## B.1. Experimental Setup of Synthetic Functions

Let input $\boldsymbol{\theta} = [\theta_i]_{i=1}^d$, the Quadratic, Cubic, Levy, and Rosenbrock functions applied in our synthetic experiments are given below:

$$F(\boldsymbol{\theta}) = \frac{1}{2} \sum_{i=1}^d \theta_i^2 \ , \tag{Quadratic}$$

$$F(\boldsymbol{\theta}) = \sum_{i=1}^d \left( |\theta_i|^3 + \frac{\theta_i^2}{2} \right) \tag{Cubic}$$

$$(76)$$

$$F(\boldsymbol{\theta}) = \sin^2(\pi w_1) + \sum_{i=2}^{d-1} (w_i - 1)^2 \left( 1 + 10 \sin^2(\pi w_i + 1) \right) + (w_d - 1)^2 \left( 1 + \sin^2(2\pi w_d) \right) \ , \tag{Levy}$$

$$F(\boldsymbol{\theta}) = \sum_{i=1}^{d-1} \left[ 100(\theta_{i+1} - \theta_i^2)^2 + (1 - \theta_i)^2 \right] \ , \tag{Rosenbrock}$$

where $w_i = 1 + \frac{\theta_i - 1}{4}$. Note that all functions have the same minimum of zero, i.e., $\min F(\boldsymbol{\theta}) = 0$. For a fair comparison, we employ the same initialization and hyperparameters: $\beta_1 = 0.9, \beta_2 = 0.99$ and $K = 10, \eta = 0.001, \mu = 0.005$, for all methods.

## B.2. Experimental Setup of Black-Box Adversarial Attack

For the black-box adversarial attack experiment on the MNIST dataset, we use the same fully trained deep neural networks from (Shu et al., 2023) and adopt a $L_\infty$ constraint of 0.2 on the input perturbation. For a fair comparison, we employ the same hyperparameters: $\beta_1 = 0.9, \beta_2 = 0.99$ and $K = 2, \eta = 0.01, \mu = 0.005$, for all methods.

## B.3. Experimental Setup of Memory-Efficient LLM Fine-Tuning

For the memory-efficient LLM fine-tuning on OPT-1.3B and OPT-13B on SST-2 dataset (Socher et al., 2013), we adopt the same configurations in (Malladi et al., 2023) and choose to fine-tune LLMs with LoRA adapters. For a fair comparison, we employ the same hyperparameters: $\beta_1 = 0.9, \beta_2 = 0.99$ and $K = 1, \eta = 0.00005, \mu = 0.001$, for all methods.

