# OpenReview forum: "Refining Adaptive Zeroth-Order Optimization at Ease"
_ICML.cc/2025/Conference — ICML 2025 poster_

### Official Review · Reviewer_MyoC · 2025-03-08

**Overall Recommendation:** 3

**Summary:**

The paper introduces a zero-order optimization technique called R-AdaZO. Specifically, it makes a simple yet seemingly effective modification to ZO-AdaMM. By replacing the second moment estimation term $g_t$ in ZO-AdaMM with $m_t$, R-AdaZO achieves variance reduction compared to ZO-AdaMM and demonstrates a faster convergence rate. The theoretical analysis is thorough.

**Claims And Evidence:**

The authors provide theoretical analysis to prove the claimed variance reduction effect. During my review, I did not identify any obvious issues (though the validation was not extremely detailed).

**Essential References Not Discussed:**

In my opinion, while ZO-AdaMM is an important baseline for this paper, the authors should provide a more detailed comparison with other common zero-order optimization (ZO) methods to demonstrate the effectiveness of their approach. This includes methods like ZO-SGD (already cited by the authors), as well as ZO-SignSGD [1] and ReLIZO [2].

[1] Liu S, Chen P Y, Chen X, et al. SignSGD via Zeroth-Order Oracle. In International Conference on Learning Representations, 2019.

[2] Wang X, Qin X, Yang X, et al. ReLIZO: Sample Reusable Linear Interpolation-based Zeroth-Order Optimization. Advances in Neural Information Processing Systems, 2024, 37: 15070-15096.

**Experimental Designs Or Analyses:**

Yes, the experiments conducted (synthetic function optimization, image black-box adversarial attacks, and LLM fine-tuning) are common application areas for zero-order optimization.

**Methods And Evaluation Criteria:**

Yes, the proposed method aims to improve the convergence speed of zero-order optimization, and the experiments conducted are reasonable.

**Other Comments Or Suggestions:**

None

**Other Strengths And Weaknesses:**

**Strengths**:
1. The paper introduces a simple yet effective modification to ZO-AdaMM, accelerating the convergence of zero-order optimization (ZO) through variance reduction.
2. The theoretical analysis is comprehensive.
3. Due to the simplicity of the proposed method, I conducted my own tests on functions like Rosenbrock. With smaller parameter dimensions (i.e., 20, 50), the proposed method shows acceleration over ZO-AdaMM. (Although, after a longer number of iterations, it may perform worse than ZO-AdaMM, likely due to the bias introduced as analyzed by the authors.) In such cases, as the parameter dimension increases, the proposed method demonstrates better scalability, as evidenced by the experiments in the paper.

**Weaknesses**:
1. I believe the authors should include comparisons with additional ZO methods to further illustrate the effectiveness of their approach. See "Essential References Not Discussed." The current experiments only compare the baseline ZO-AdaMM and the ZO-RMSprop, which is essentially similar to ZO-AdaMM.

**Questions For Authors:**

1. See weaknesses. I am curious about the performance of other ZO methods under the same experimental setup used by the authors.
2. Could you provide a detailed performance comparison between R-AdaZO and ZO-AdaMM with longer iterations (where both methods converge)? For example, using the synthetic function in Figure 1.

**Relation To Broader Scientific Literature:**

The paper introduces a new zero-order optimization algorithm based on ZO-AdaMM, aiming to improve the convergence speed of zero-order optimization methods. This is largely consistent with previous approaches in the field.

**Theoretical Claims:**

Yes, I verified some of the theoretical proofs. The assumptions and lemmas used are consistent with prior work in the zero-order optimization (ZO) field, and I did not find obvious issues.

---

> ### Author Rebuttal · Authors · 2025-04-01
>
> We are grateful to Reviewer MyoC for the positive and constructive feedback! We appreciate that the reviewer highly recognizes that our paper is **simple yet effective** and our theoretical analysis is **comprehensive**. We would like to address your comments as follows.
>
> ---
>
> > ...the authors should provide a more detailed comparison with other common zero-order optimization (ZO) methods...
>
> We appreciate the reviewer's suggestion to provide a more detailed comparison with common zero-order optimization (ZO) methods. Following this recommendation, we have benchmarked our R-AdaZO against ZO-SGD and ZO-signSGD on the Quadratic and Levy functions, using the experimental setup detailed in Section 6.1.
>
> **Quadratic Function Performance:**
> | Quadratic        | $T=1 \times 10^4$ | $T=2\times10^4$ | $T=3\times10^4$ | $T=4\times10^4$ | $T=5\times10^4$ |
> |---------|--------|---------|---------|---------|---------|
> |ZO-SGD| 5038.056|5028.611|5019.293|5009.897|4997.447 |
> |ZO-signSGD| 3644.077|1983.347|834.561|190.489|2.159 |
> |**R-AdaZO**| **756.743**|**0.599**|**0.029**|**0.030**|**0.030** |
>
> **Levy Function Performance:**
> | Levy       | $T=1 \times 10^4$ | $T=2\times10^4$ | $T=3\times10^4$ | $T=4\times10^4$ | $T=5\times10^4$ |
> |---------|--------|---------|---------|---------|---------|
> |ZO-SGD| 6661.504|6625.426|6589.794|6553.917|6506.384 |
> |ZO-signSGD| 4017.191|1579.627|779.139|606.901|562.197 |
> |**R-AdaZO**| **795.038**|**575.398**|**560.107**|**558.741**|**558.716** |
>
> As demonstrated in the tables, R-AdaZO consistently and significantly outperforms both ZO-SGD and ZO-signSGD. Regarding the ReLIZO approach: While ReLIZO presents a valuable technique for enhancing *ZO gradient estimation*, our current work's primary focus is on improving the *ZO update mechanism*. Integrating R-AdaZO with advanced gradient estimators like ReLIZO is an interesting direction for future research, which we plan to explore subsequently. We believe maintaining this focus clarifies the specific contributions of R-AdaZO in this submission.
>
> > Could you provide a detailed performance comparison between R-AdaZO and ZO-AdaMM with longer iterations (where both methods converge)?
>
> Thank you for this insightful question regarding the long-term convergence behavior. To address this, we have extended the comparison between R-AdaZO and ZO-AdaMM, running the experiments for 6 times the number of iterations used in the main paper, maintaining the settings from Section 6.1.
>
> **Quadratic Function Performance (Extended Iterations):**
> | Quadratic        | $T=37500$|$T=41250$|$T=45000$|$T=48750$|$T=52500$|$T=56250$|$T=60000$ |
> |---------|--------|---------|---------|---------|---------|---------|---------|
> |ZO-AdaMM | 28.972|2.509|0.101|0.027|0.026|0.026|0.026 |
> |R-AdaZO| 0.030|0.030|0.030|0.029|0.030|0.030|0.030 |
>
> **Levy Function Performance (Extended Iterations):**
> | Levy   | $T=37500$|$T=41250$|$T=45000$|$T=48750$|$T=52500$|$T=56250$|$T=60000$ |
> |---------|--------|---------|---------|---------|---------|---------|---------|
> |ZO-AdaMM | 584.637|572.480|565.976|562.638|561.050|560.467|560.252 |
> |R-AdaZO| 557.904|557.821|557.816|557.824|557.824|557.823|557.811 |
>
> As shown in the results above, even with significantly more iterations, R-AdaZO achieves a final performance level that is comparable to, and in the case of the Levy function, slightly better than that of ZO-AdaMM.
>
> ---
>
> We thank the reviewer for the valuable input and we will incorporate these discussions into our revised version. We hope our clarification will address your concerns and improve your opinion of our work.

---

> > ### Comment · Reviewer_MyoC · 2025-04-03
> >
> > Thank you for the detailed response.  The performance results of R-AdaZO and ZO-AdaMM with longer iterations provided by the authors are in line with my expectations. That is, after convergence, the final difference between the two methods is small, but R-AdaZO demonstrates a faster convergence rate, which will significantly enhance its scalability.
> >
> > I appreciate the comparison results with ZO-SGD and ZO-SignSGD provided by the authors (which should be included in the experimental section or the appendix), and I also agree with the authors' discussion on the differences with ReLIZO (which should be included in the related work or future work).
> >
> > Overall, I believe the paper presents a simple yet effective improvement, and I am inclined to recommend acceptance.

---

> > > ### Author Response · Authors · 2025-04-03
> > >
> > > Dear Reviewer MyoC,
> > >
> > > We are very happy that our response has addressed your concerns and improved your opinion about our work! We will include those valuable discussions and additional results in the final version.
> > >
> > > Best regards,
> > > Authors

---

### Official Review · Reviewer_6KdC · 2025-03-10

**Overall Recommendation:** 5

**Summary:**

The authors take a closer look at the Adam update rule in the context of Zero-Order Neural Network training and identify a specific change to the second-moment update rule that provides improved convergence theoretically and empirically.
Their experimental sections contains synthetic experiments as well as neural-network based experiments.

**Claims And Evidence:**

The authors claim a new update rule for ZO optimization, the use-fullness of which is well supported.
They further claim new theoretical insights, which I didn't parse completely. I find formulations such as "we theoretically show that [...] is likely to be [...] and may better [...]" to be problematic. I would suggest to clearly separate what is theoretically shown versus what is speculation.
The final claim concerns experimental validation, which is convincing.

**Essential References Not Discussed:**

No

**Experimental Designs Or Analyses:**

Yes. I see no obvious issues.

**Methods And Evaluation Criteria:**

Yes

**Other Comments Or Suggestions:**

Line 284 left: "to decompose"

**Other Strengths And Weaknesses:**

Many good papers stem from simple ideas, such as this one. I see no obvious weakness.

**Questions For Authors:**

While the experimental evaluation is varied, it is not very extensive.
I'd love to see a more detailed evaluation across more elements from the ZO-Bench suite (https://github.com/ZO-Bench/ZO-LLM) than OPT on SST2. In addition, theoretical analysis showcased a significant influence of $\beta_1$. It would be interesting to see if that is reflected in the convergence of LLM fine-tuning. Should the authors expand on their experimental evaluation as mentioned, I will consider raising my score.

**Relation To Broader Scientific Literature:**

The wider scientific literature on ZO is focussed on variance-reduction through various theoretical and empirical observations and improvements. This paper specifically is orthogonal to many of these papers and focusses on a single straight-forward change to the Adam update rule.

**Theoretical Claims:**

I followed the theoretical claims of section 5.1 and 5.1, but did not check 5.3.
I did not check the proofs in the appendix.

---

> ### Author Rebuttal · Authors · 2025-04-01
>
> We are grateful to Reviewer 6KdC for the positive assessment and valuable feedback! We particularly appreciate the recognition that "Many good papers stem from simple ideas, such as this one". We would like to address your concerns below.
>
> ---
> > I find formulations such as "we theoretically show that [...] is likely to be [...] and may better [...]" to be problematic. I would suggest to clearly separate what is theoretically shown versus what is speculation
>
> Thank you for this crucial suggestion regarding the precision of our claims. We agree completely. In the revised manuscript, we will meticulously differentiate between:
> - Formally proven theoretical results: Statements directly supported by theorems and proofs within the paper.
> - Heuristic arguments or intuitions: Explanations for why certain phenomena might occur, clearly marked as such.
> - Empirical observations: Findings derived solely from experimental results.
>
> We will ensure the language clearly reflects these distinctions throughout the paper for improved accuracy and clarity.
>
> > I'd love to see a more detailed evaluation across more elements from the ZO-Bench suite (https://github.com/ZO-Bench/ZO-LLM) than OPT on SST2.
>
> Following your suggestion, we have expanded our evaluation using the ZO-Bench suite. We benchmarked R-AdaZO against ZO-AdaMM on the Copa dataset using the OPT-13B model, adhering to the ZO-Bench setup. The results (loss values) are presented below:
>
> **Performance on Copa (OPT-13B):**
> | Copa   | $T=1 \times 10^3$ |$T=2 \times 10^3$|$T=3 \times 10^3$|$T=4 \times 10^3$|$T=5 \times 10^3$|$T=6 \times 10^3$|$T=7 \times 10^3$|$T=8 \times 10^3$|$T=9 \times 10^3$|$T=10 \times 10^3$|
> |---------|--------|---------|---------|---------|---------|---------|---------|---------|---------|---------|
> |ZO-AdaMM | 2.527|2.352|2.133|2.004|1.938|1.636|1.579|1.716|1.529|1.696 |
> |R-AdaZO| 2.527|2.051|1.521|1.584|1.482|**1.297**|1.420|1.531|1.405|1.640 |
>
> These results demonstrate that R-AdaZO exhibits significantly faster convergence compared to ZO-AdaMM on this extended LLM fine-tuning task. This provides further evidence for the effectiveness of R-AdaZO on diverse tasks within the ZO-Bench framework.
>
> > In addition, theoretical analysis showcased a significant influence of $\beta_1$. It would be interesting to see if that is reflected in the convergence of LLM fine-tuning.
>
> Thank you for pointing out this connection between our theory and empirical results. To investigate this, we performed an ablation study on the effect of $\beta_1$ for R-AdaZO during LLM fine-tuning. We used the SST2 dataset with the OPT-1.3B model, following the setup in Section 6.3, varying only $\beta_1$.
>
> **Performance on SST2 (OPT-1.3B) with varying $\beta_1$ for R-AdaZO:**
> | SST2   | $T=1 \times 10^3$ |$T=2 \times 10^3$|$T=3 \times 10^3$|$T=4 \times 10^3$|$T=5 \times 10^3$|$T=6 \times 10^3$|$T=7 \times 10^3$|$T=8 \times 10^3$|$T=9 \times 10^3$|$T=10 \times 10^3$|
> |---------|--------|---------|---------|---------|---------|---------|---------|---------|---------|---------|
> |$\beta_1=0.3$ | 1.063|0.736|0.490|0.612|0.414|0.429|0.389|0.265|0.276|0.187 |
> |$\beta_1=0.6$ | 1.063|0.675|0.387|0.493|0.330|0.360|0.388|0.232|0.282|0.229 |
> |$\beta_1=0.9$ | 1.063|0.462|0.178|0.289|0.165|0.329|0.251|0.138|0.255|0.161 |
>
> The results clearly show that the choice of $\beta_1$ significantly impacts performance, empirically validating the implications of our theoretical analysis in the context of LLM fine-tuning. Specifically, a larger value of $\beta_1=0.9$ consistently leads to faster convergence and achieves lower loss values throughout the fine-tuning process compared to smaller values ($\beta_1=0.3,0.6$). This confirms that the theoretical insights regarding $\beta_1$ translate to practical benefits in this application domain.
>
> ---
> Thank you again for the valuable feedback. We will integrate the discussed clarifications and new experimental findings into the revision. We hope our clarification will address your concerns and further improve your opinion of our work.

---

### Official Review · Reviewer_FBQc · 2025-03-12

**Overall Recommendation:** 3

**Summary:**

This paper proposes an improvement over existing adaptive-gradient methods for zeroth-order optimization (ZO).
In ZO optimization, to minimize a function $\min_{\theta \in \mathbb{R}^d} F(\theta)$ where $F(\theta) = \mathbb{E} [ f (\theta;\xi)] $, we can only access noisy function values $f(\theta;\xi)$. ZO methods use a finite difference of functions as an approximation of the gradient : $\hat{\nabla}f(\theta;\xi) \triangleq \frac{d}{K}\sum_{k=1}^K \frac{f(\theta +\mu u_k;\xi) - f(\theta;\xi)}{\mu}u_k$, where $u_k\overset{iid}{\sim}Unif(\mathbb{S}^{d-1})$ and $\mathbb{S}^{d-1}$ is the $d$-dimensional sphere with a fixed $\mu>0$.

ZO methods suffer with high noise variance for their estimated gradients especially for high dimensions $d\gg K$. And this

1. The authors claim that momentum methods, however, offer some sort of "automatic variance-reduction" to handle this.


2. Existing baselines which use momentum, for instance the ZO-AdaMM(Chen et al 2019), that uses  $\hat{\nabla}f(\theta;\xi)$, instead of the true gradient in Adam, however, do not take advantage of this fact. The authors propose a small modification of ZO-AdaMM, to get R-AdaZO(Algorithm 2), that estimates the second moment as $v_t \gets \beta_2 v_{t-1} + (1 - \beta_2)m_t^2$, where $m_t$ is the momentum term. Note that AdaMM uses, $v_t \gets \beta_2 v_{t-1} + (1-\beta_2)g_t^2$, where $g_t$ is the ZO estimate of gradient.

3. **Theory**:
    1. The authors show that estimates of second moment $v$ are smaller for R-AdaZO than ZO-AdaMM by incorporating the variance reduction due to momentum (Theorem 5.4 and Corollary 5.5).
    2. The authors use a "variance-aware" convergence analysis for non-convex $F$, by bounding norm of $\nabla F_{\mu}(\theta)$ where $F_{\mu}(\theta) = \mathbb{E}_{u\sim Unif(\mathbb{S}^{d-1})}[F(\theta + \mu u)]$ is the appropriate objective for ZO methods. They show that existing analysis of ZO-AdaMM is limited as it considers only one term of the upper bound on gradient norm obtained by a Holder's inequality in Lemma 5.6.
    3. In Theorem 5.9 and Corollary 5.10, they show that their method obtains a strictly smaller error in gradient norm than ZO-AdaMM with the difference being an additive term of $\sqrt{V}$ for R-AdaZO v/s $\sqrt{\hat{V}}$ for AdaMM, with $\sqrt{\frac{V}{\hat{V}}} = \mathcal{O}(\sqrt[4]{\frac{1-\beta_1}{1+\beta_1}})$.

4. **Experiments**:
    1. For optimizing synthetic loss functions(Fig 1), fine-tuning an LLM (Fig 3) or performing a black-box adversarial attack (Table 1) their method is significantly faster than baselines AdaMM and RMSProp with ZO gradient estimates.
    2. In Figure 2, they show that the error in first moment $m_t$ and second moment $v_t$ wrt true gradients is much smaller for their method. They also show its behavior for different $\beta_1$.

**Claims And Evidence:**

Several theoretical claims are justified by corresponding Theorems and Corollaries mentioned in the Summary as well the Remarks following them. The theoretical claims that are problematic or not justified are explained below --

1. **Incomplete proof for Corollary 5.10**: Corollary 5.10 provides the performance of the baseline ZO-AdaMM under the authors' proposed variance-aware convergence analysis. The authors do not provide its proof and claim that it can be obtained by simply following the proof of Theorem 5.9, which is their method R-AdaZO. I do not think this is the case, as several steps in proof of Theorem 5.9 utilize the exact update of R-AdaZO and might not work for ZO-AdaMM with a different update. These steps are  -- i) Eqn. (46), which is the main bound in the descent Lemma, specifically terms $2$ and $3$, ii) Eq. (56) and (57) which basically bound the convergence rate for $T$ update steps. These are important steps in the proof of R-AdaZO and without any explanation of how these would be addressed for the baseline ZO-AdaMM, it is unclear if the claimed rate in Corollary 5.10 is actually achieved for ZO-AdaMM, and if it is indeed worse theoretically than R-AdaZO. This is one of the main reasons for my rejection, and I would be willing to change my assessment if the authors can provide this proof.

2. **Justification of Assumptions** The assumption that the function values are always bounded in Assumption 1 is extremely strong. Several works assume a lower bound on the function value, but assuming an upper bound on the function forces some form of subgaussian tails on it. If the domain of models $\theta$ was a closed compact subset $\Theta$ of $\mathbb{R}^d$, then this assumption could be justified. Even then, the authors would need to specify the value of $\mu$ so that any noisy evaluation $\theta + \mu u$ is not outside $\Theta$. Note that none of the synthetic functions used in their experiments except the rastrigin function are unbounded for  $\theta \in \mathbb{R}^d$. Note that this also doesn't hold for a quadratic function.


3. **Final Convergence Rate** Assuming that Corollary 5.10 is actually correct and the authors provide a proof in their rebuttal, it still does not imply a speed-up in terms of the final convergence rates for R-AdaZO compared to ZO-AdaMM theoretically, as the error terms where these two algorithms differ might not be the dominant term in error. The authors should thus write down precise conditions when this does not happen, in terms of $d, \beta_1, \beta_2,\zeta,\mu$ and the initialization $m_0, v_0$. Further, the authors should check what is the maximum possible speedup possible in convergence rate in terms of $\epsilon$ instead of keeping the speedup in terms of $\beta_1$ for only additive terms? Note that in the current form in Remark on Page 7, R-AdaZO does not denote a speed-up as long as the terms of $\hat{V}$ and $V$ are not the dominant error terms in Theorem 5.9 and Corollary 5.10, as only then do they actually decide the convergence speed.


**Experiments**: Note that the experimental performance for their method is significantly better than baselines for both synthetic and real tasks as demonstrated in Figures 1 and 3.

**Essential References Not Discussed:**

I think most essential references were discussed. They have a  showed that the variance-reduction was observed originally in (Liu et al 2020), but they were the first ones to use it for ZO updates. Further, they discussed the main baseline for adaptive ZO optimization : ZO-AdaMM (Chen et al 2019). They missed a few old references on the smoothing, which is exactly ZO optimization but the analysis utilizes smoothness of the resultant function $F_\mu$ (Nesterov 2005, Duchi et al 2012). Note that the "variance-aware convergence analysis" mentioned in Section 5.3 seems to be analysis in terms of the smoothed objective $F_{\mu}$ which has been performed in the references I mentioned, although not for Adam-style adaptive algorithms.

**References**
- (Nesterov 2005) Smooth minimization of non-smooth functions. Math Programming.
- (Duchi et al 2012) Randomized Smoothing for Stochastic Optimization. SIAM J. OPT.

**Experimental Designs Or Analyses:**

Yes, I checked all the experimental designs and analyses.

**Methods And Evaluation Criteria:**

The experiments on synthetic data and finetuning had all the sufficient details.

- **Is cosine similarity a good metric in Fig 2(a)?**: In Fig 2(a), the authors compare the cosine similarity between actual gradient $\nabla f(\theta)$ and the momentum estimate $m$ from ZO-AdaMM and R-AdaZO. In Theorem 5.3, they bound the expected squared error coordinate-wise for this quantity, so measuring cosine similarity instead of squared difference does not seem appropriate. Further, for both Fig 2(a) and (b), we don't know if the error is measured using momentum at a specific time of the algorithm or averaged over the algorithm's trajectory. This detail is important for momentum as it might not have kicked in early in the algorithm and saturated if measured very late.


- **No justification for black-box attack experiments**: I don't understand the reason for testing an optimization algorithm on black-box attack. The authors only state "Following the practice in (Shu et al 2023)" in Line 367 2nd column as the justification. In (Shu et al 2023), I would the actual reason for this, as black-box attacks are an important application for ZO optimization, as an adversary can only access the function values during attack. I think the authors should mention this justification in their paper as well.

**Other Comments Or Suggestions:**

- **Writing**: Some parts of the writing can be improved. For instance, there is some repetition of the main claims verbatim atleast $4-5$ times in the paper. See the following typos --
    1. The unit sphere in $d$ dimensions is referred to as $\mathbb{S}$ instead of the standard $\mathbb{S}^{d-1}$. If non-standard notation is used, it should be specified in the problem setup (Section 3).
    2. The exact formulation of optimality gap (in terms of either gradients, function values or iterate distance) in the synthetic experiments was not specified.
    3. Line 919 RHS of first equation: $\frac{1}{\sqrt{\beta_2}v_{0,i}+\zeta}$ -> $\frac{1}{\sqrt{\beta_2 v_{0,i} + \zeta}}$.
    4. Line 858: $\frac{|m_{t,i}|}{\sqrt{1-\beta_2}|m_{t+1,i}|}$ -> $\frac{|m_{t,i}|}{\sqrt{1-\beta_2}|m_{t,i}|}$.
    5. Line 587 : Missing reference -> Eq (16).

Overall, I am leaning towards rejection due to the main issues in the "Claims and Evidence" section. I'm willing to change my assessment if they can address them.

**Other Strengths And Weaknesses:**

1. **Dependence on dimension $d$**: Note that the authors have not explicitly characterized the optimal dependence on the dimension $d$ in Theorems 5.9 and Corollary 5.10. Note that the noise variance in ZO is high because it grows with $d$. In adaptive Adam-style optimization, the goal is to use different learning rates for different coordinates and the analysis is on a per-coordinate basis. Therefore, for ZO in Adam-style algorithms, the dependence of error on dimension $d$ is extremely important. Further, several steps of the analysis involve changing norms from coordinate-wise to $\ell_2$, so I suspect the dimension dependence is not good. Note that the error in baseline ZO-AdaMM (Chen et al 2019) for the term $B$ in Lemma 5.6 was $\frac{\sqrt{d}}{\sqrt{T}}$, thus giving a $\frac{d}{\epsilon^2}$ error bound. Can the authors provide sharp bounds in terms of $d$ as well?




2. **Extension to gaussian smoothing distribution**: The authors sample the random vectors for gradient estimate from a uniform distribution on the sphere. Instead if the random vectors were sampled from a gaussian distribution, which in high dimensions is extremely similar to a uniform distribution, the proofs would not go through as several terms bounds won't hold, for instance Eq (50). Is there an easy fix for this gaussian extension or are the methods designed specifically for uniform distribution on the sphere with bounded function values (Assumption 1).

**Questions For Authors:**

- How do the proposed methods perform in comparison to other ZO methods that keep track of the history instead (Shu et al 2023)? Are both these techniques completely orthogonal, so that they can be combined to obtain a "variance-reduced" ZO that uses historical estimates?
- As a follow-up for the previous question, are there minmax lower bounds in terms of number of function evaluations for this ZO model, like (Duchi et al 2015)? These could tell us if the obtained method is optimal or if there is room for improvement?
- Why is there a mismatch between theory and practice in Figure 2(b) for large $\beta_1$? Can you further expand on why the loose bound is still good for convergence analysis?



**References**--
- (Duchi et al 2015) Optimal Rates for Zero-Order Convex Optimization: The Power of Two Function Evaluations. Trans on Info. Theory.

**Relation To Broader Scientific Literature:**

Crucially, the authors use the variance-reduction properties of momentum first observed in (Liu et al 2020) for first-order optimization, i.e., with access to gradients. Using this insight for the problem of adaptive ZO optimization, shows that simply applying ZO gradient estimates to Adam is suboptimal and the small change in R-AdaZO actually results in better convergence.

**Theoretical Claims:**

Yes, I checked all the proofs.
Apart from the issues mentioned in "Claims and Evidences" section, there are several minor mistakes in the proof. These errors are mostly in explanations not in the equations and they don't break the proof.
- Lines 561 and 562 should have $\leq$ instead of $=$ as a Cauchy-Schwarz inequality is used.
- Line 927 : The authors used $\frac{a}{1+a} \leq \ln(1+a)$ in Eq (56) and (57) but wrote down the inequality $\ln(1+a) \leq a$.
- Line 963 : The inequality should be $\sum a_i/b_i \geq  \sum_i a_i/\sum_i b_i$ for $b_i \geq 0$, which is what the authors actually used. They mentioned the inequality in the opposite direction.

---

> ### Author Rebuttal · Authors · 2025-04-01
>
> We sincerely thank the reviewer for their diligent, detailed, and constructive feedback, and appreciate the time and effort invested . We found the comments very helpful and address each point below.
>
> ---
> > Incomplete proof
>
> Thank you for highlighting the need for clarity regarding Corollary 5.10 (ZO-AdaMM). You are correct; its proof requires adaptation from Theorem 5.9 due to the different second moment update rule ($v_t$). While our overall variance-aware framework (Lemma 5.6, Eq. 45/46) remains applicable, the specific bounds for terms ①-④ within Eq. (46) must be re-evaluated using ZO-AdaMM's $v_t$.
> - **Terms ① (Eq. 47) & ④ (Eq. 52):** The bounds analogous to (47) and (52) hold for ZO-AdaMM.
> - **Terms ② (Eq. 48) & ③ (Eq. 51):** We can adapt **Lemma 6 from Zhang et al. (2024a)**, which provides a bound for $|m_{t,i}| / \sqrt{v_{t,i}+\zeta}$ in the context of standard Adam (where $v_t$ uses $g_t^2$). Applying this lemma (adapted for the ZO variance $\Sigma^2$) allows us to obtain bounds analogous to those derived through (48) and (51), although the constants involved will differ.
>
> For bounding the summation terms analogous to (56)/(57) in the ZO-AdaMM proof, we leverage **Lemma 7 from Zhang et al. (2024a)**, which handles sums involving $v_t$ based on $g_t^2$. Combining above, we can rigorously derive the rate in Cor 5.10. This rate depends on $\hat{V}$ (reflecting the $g_t^2$ update, Cor 5.5), confirming R-AdaZO's theoretical advantage via the smaller variance term $V$ (Thm 5.4).
>
> Due to rebuttal space limits, we provide this proof sketch; a full derivation referencing these adapted steps will be included in the appendix of the revision. Thank you again for your careful analysis.
>
> > Justification of Assumptions
>
> Regarding Assumption 1 (8): While the upper bound is strong (especially for quadratics/unbounded domains), it's a common theoretical tool (like bounded gradients in Chen et al., 2019) essential for controlling variance and deriving convergence guarantees in non-convex stochastic settings. Our theory relies on this standard simplification, but experiments confirm R-AdaZO's practical robustness where it doesn't strictly hold globally.
>
> > Final Convergence Rate
>
> You're right, the overall speedup requires the variance term ($V$ vs $\hat{V}$) to dominate. We contend this is common in ZO due to high estimation noise, making R-AdaZO's better variance handling crucial. Pinpointing exact dominance conditions is hard, but our analysis highlights the structural gain against this main ZO challenge. We'll clarify the remark on Page 7 to state the speedup applies primarily when variance dominates, matching typical ZO scenarios and our results.
>
> > Methods And Evaluation Criteria:
>
> Thank you for the feedback and suggestions. Regarding Fig 2(a), cosine similarity visualizes the improved *directional alignment* of $m_t$, complementing the squared error theory. Metrics in Fig 2 are averaged over the trajectory. We will clarify this rationale and the averaging. We also agree the black-box attack justification needs explicit motivation (its relevance as a key ZO application due to lack of gradient access for attackers) and will add it. These clarifications will be incorporated.
>
> > Theoretical Claims & Other Comments
>
> Thank you for your meticulous review and for carefully checking all the proofs. We appreciate you pointing out these minor errors in the explanations. We understand they don't affect the core correctness of the proofs, and we will certainly correct these textual issues in the revised version.
>
> > Essential References Not Discussed
>
> Thank you for the feedback and for acknowledging our discussion of key references. We appreciate you suggesting the additional references on smoothing (Nesterov 2005, Duchi et al. 2012) and will certainly incorporate them into our related work section.
>
> > Dependence on dimension $d$
>
> Thank you for highlighting dimension $d$ dependence. Our analysis uses coordinate-wise Lipschitz, standard for adaptive methods but differing from Euclidean Lipschitz (e.g., Chen et al. 2019), making direct $d$ comparison complex. While $d$ is implicit in our terms, our focus was demonstrating variance reduction and the refined second moment's benefit under this framework, not optimizing $d$ dependence. Achieving sharp $d$ bounds is a valuable, challenging direction for future research.
>
> > Extension to gaussian
>
> You are correct, our proofs leverage properties of the uniform sphere estimator simplifying bounds like Eq. (50). Gaussian smoothing's different statistics require adapted tools and assumptions (e.g., concentration inequalities) for analysis. Adapting our framework needs re-derivation, not a quick fix, though R-AdaZO's core principles should still apply.
>
> > Questions
>
> All the remaining questions will be addressed in our revised version due to space limit.
>
> ---
> With our elaboration, we hope our response has addressed your concerns and increased your opinions of our work.

---

> > ### Comment · Reviewer_FBQc · 2025-04-03
> >
> > Thank you for the detailed response.
> >
> > I'm increasing my score as I'm convinced that Corollary 5.10 can possibly be proved using Lemmas 6 and 7 from (Zhang et al 2024a).
> >
> > Is there a reference that justifies the statement that the term variance dominates in zeroth-order optimization? I understand that the empirical results here somehow demonstrate this, but is there a mention of this in any existing ZO works theoretically? Or is it observed in practice for most cases, not just the Adam-style algorithms described here?

---

> > > ### Author Response · Authors · 2025-04-03
> > >
> > > Dear Reviewer FBQc,
> > >
> > > Thank you very much! We are glad our response helped address your concerns and improve your assessment of our work. We will certainly incorporate these valuable discussions into the final version.
> > >
> > > Regarding your question about the dominance of variance in ZO optimization:
> > >
> > > Our theoretical analysis provides direct justification via **Eq. (15)**. This equation explicitly shows that the variance of the ZO gradient estimator ($\Sigma^2$) inherently scales with dimension $d$ and inversely with smoothing parameters ($\mu^2, K$). This structure makes $\Sigma^2$ fundamentally larger than typical first-order variance ($\sigma^2$), particularly in the high-dimensional settings where ZO methods are often applied.
> > >
> > > This large, dimension-dependent variance is widely recognized as a central challenge in the ZO literature, e.g., (Liu et al., 2018a). Consequently, it often becomes the dominant factor limiting the overall convergence rate compared to other terms like bias or optimization error, especially as $d$ grows. While formal proofs of dominance across *all* possible parameter regimes might be intricate, tackling this high variance is a primary focus of much ZO research.
> > >
> > > Our empirical results strongly support this view:
> > > 1.  The significant performance gains shown in Sec. 6 stem directly from R-AdaZO's improved handling of variance.
> > > 2.  Our preliminary results (which we will add) show that the same refinement (Eq. 6) offers minimal benefit in low-variance first-order settings, highlighting that its advantage is most pronounced when variance *is* the bottleneck, as in ZO.
> > >
> > > We hope this clarifies why variance is considered a dominant factor in ZO optimization, supported by both theoretical structure and empirical evidence.
> > >
> > > Best regards,
> > >
> > > Authors

---

### Decision · Program_Chairs · 2025-05-01

**Decision:**

Accept (poster)

**Comment:**

This paper proposes R-AdaZO, a refined adaptive zeroth-order optimization method. R-AdaZO exploits the variance reduction effect of the first moment estimate to enhance the accuracy and stability of gradient-free updates and refines the second moment estimate to better capture the geometry of the optimization landscape. Theoretical analysis provides variance-aware convergence guarantees and demonstrates faster convergence compared to existing methods. Extensive experiments on synthetic tasks, black-box adversarial attacks, and memory-efficient fine-tuning of large language models confirm the practical effectiveness of R-AdaZO.

All reviewers acknowledged the soundness of the proposed techniques, the theoretical analysis, and the strength of the experimental results. One reviewer raised concerns about aspects of the theoretical analysis, including restrictive assumptions and suboptimal dependence on the dimensionality d. Another two reviewers noted the absence of evaluations on the ZO-Bench suite and the limited number of baselines. The authors have responded to these concerns in the rebuttal by providing additional analysis and experiments.

Given the overall contributions and the authors’ responsiveness in addressing key concerns, we believe the strengths of the submission outweigh its limitations. We therefore recommend acceptance, and encourage the authors to revise the manuscript in line with the reviewers’ feedback and their rebuttal commitments.